# TBL1X/TBL1XR1 govern β-cell identity through a PAX6-containing gene regulatory network

Alina A. Walth-Hummel[1,2,3], Celine Jouffe[1,3], Peter Weber[1,2], Karsten Motzler[1,2,3], Julia Geppert[1,2], Michael Sterr[3,4], Wei Gan[5], Iwona Szczerbinska[6], Ann-Christine König[7], Stefanie M. Hauck[3,7], Raul Terron-Exposito[1,2], Daniela Hass[1,2], Congcong Wang[1,3,8], Kenneth A. Dyar[1,3], James G. Lyon[9], Heiko Lickert[3,4,8], Carina Ämmälä[6], Stephan Herzig[1,2,3,10], Frances M. Ashcroft[11], Mostafa Bakhti[3,4], Patrick E. MacDonald[9] & Maria Rohm[1,2,3,12] ✉

A main mechanism of β-cell dysfunction in diabetes is loss of identity, controlled by transcription factors that induce identity gene expression and disallowed gene repression. How transcription factors facilitate simultaneous expression and repression is not fully understood, representing a knowledge gap in diabetes research. We identify the transcriptional co-factors transducin β-like 1 x-linked (TBL1X) and its homolog TBL1X-related (TBL1XR1, together TBL/R1) as crucial regulators of β-cell identity and determinants of diabetes development and progression. β-cell specific TBL/R1 knockout in mice leads to progressive hypoinsulinemia and hyperglycemia. scRNA-sequencing reveals loss of β-cells, emergence of polyhormonal cells, and reduced β-cell maturity upon TBL/R1 knockout. Interactome screens and chromatin immunoprecipitation show TBL/R1 directly regulate insulin promoter activity through a PAX6-HDAC3 gene regulatory network, evident also in human models. TBL/R1 associates with diabetes in humans, thus our study uncovers an additional regulatory layer maintaining β-cell identity crucial for diabetes development and progression.

Diabetes is a worldwide health threat mainly resulting from the pancreatic β-cells' inability to produce sufficient insulin relative to insulin requirements. Insulin synthesis and secretion are maintained by pancreatic β-cells. A decline in functional β-cell mass precedes type 2 diabetes mellitus (T2DM) and is causative for the progressive nature of the disease.

β-cell decline in T2DM is thought to occur primarily as a consequence of marked changes in the β-cell phenotype, such as the reduced expression of insulin and the loss of β-cell identity[1–4], both of which are further amplified by hyperglycemia[5,6]. β-cell identity is maintained by a series of transcriptional events governed by well-established transcription factors, including pancreatic and duodenal

[1]Institute for Diabetes and Cancer (IDC), Helmholtz Diabetes Center, Helmholtz Center Munich, Neuherberg, Germany. [2]Joint Heidelberg-IDC Translational Diabetes Program, Inner Medicine 1, University Hospital, Heidelberg, Germany. [3]DZD (German Center for Diabetes Research), Munich, Germany. [4]Institute of Diabetes and Regeneration Research (IDR), Helmholtz Diabetes Center, Helmholtz Center Munich, Neuherberg, Germany. [5]Human Genetics Centre of Excellence, Novo Nordisk Research Centre Oxford, Oxford, UK. [6]Department of Discovery Biology and Pharmacology, Novo Nordisk Research Centre Oxford, Oxford, UK. [7]Metabolomics and Proteomics Core (MPC), Helmholtz Center Munich, Neuherberg, Germany. [8]School of Medicine and Health, Technical University of Munich, Munich, Germany. [9]Department of Pharmacology and Alberta Diabetes Institute, University of Alberta, Edmonton, Canada. [10]Chair Molecular Metabolic Control, Technical University Munich, Munich, Germany. [11]Department of Physiology Anatomy and Genetics, University of Oxford, Oxford, United Kingdom. [12]DZHK (German Center for Cardiovascular Research), partner site Munich Heart Alliance, Munich, Germany. ✉e-mail: maria.rohm@helmholtz-munich.de

homeobox 1 (PDX1), NK6 homeobox 1 (NKX6.1), or paired box protein 6 (PAX6)[7–9]. These regulate the expression of β-cell genes (e.g., insulin (*INS* in humans, *Ins1* and *Ins2* in rodents), MAF bZIP transcription factor A (*MAFA/Mafa*), the glucose transporter solute carrier family 2 member 2 (*SLC2A2/Slc2a2*))[7–11], while repressing the expression of non-β-cell genes, so-called disallowed genes (e.g., hexokinase 1 (*HK1/Hk1*), lactate dehydrogenase a (*LDHA/Ldha*)), and non-β-cell hormones (e.g., glucagon (*GCG/Gcg*))[7,9,12,13].

The upregulation of disallowed and progenitor genes causes β-cells to exit their mature and differentiated state, leading to dysfunctional insulin secretion, which promotes T2DM. The reversibility of the progressive loss of β-cell identity e.g., upon diabetes remission[5,14] highlights the need for an in depth understanding of β-cell identity determination and maintenance. Specifically, how exactly β-cell transcription factors function simultaneously as activators and repressors within the same cell is an important open question in the field[11].

One regulatory layer required for fine-tuning islet transcription factor activity is their interaction with transcriptional co-factors to maintain functional gene networks[15,16]. The functional gene network of most β-cell transcription factors, including PAX6, still remains to be discovered. Transducin β-like 1 x-linked (TBL1X) and TBL1X-related (TBL1XR1) (hereafter referred to as TBL/R1) are transcriptional co-factors involved in mediating both repression and activation of transcription[17,18]. Initially identified as core components of the multi-protein repressor complex nuclear receptor corepressor/silencing mediator of retinoic acid and thyroid hormone receptors (NCOR/SMRT), together with histone deacetylase 3 (HDAC3) and G-protein pathway suppressor 2 (GPS2)[19,20], TBL/R1 modulate the activity of transcription factors such as nuclear receptors, β-catenin, or nuclear factor kappa B (NFκB)[18,21,22].

TBL/R1 function is diverse as they influence transcriptional events by altering transcription factor activity and chromatin accessibility, which is mediated through differential binding of co-factors such as ubiquitin ligases, methyltransferases, or acetyltransferases/ deacetylases. We and others have demonstrated that TBL/R1 critically regulate metabolism in tissues relevant for diabetes, including adipose tissue and liver, through tissue-specific gene networks[17,23,24], hence TBL/R1 gene regulatory networks seemed to have wider implications for diabetes development. Their role in β-cells has yet to be defined.

Here, we show that TBL/R1 are required for the maintenance of functional β-cells in mice. Two independent mouse models of β-cell-specific TBL/R1 knockout display progressive hyperglycemia due to the β-cell's inability to synthesize sufficient insulin. Bulk and single-cell RNA-sequencing show that β-cell gene signatures are lost in pancreatic islets upon β-cell TBL/R1 knockout, characterized by a reduced expression of β-cell identity genes and increased expression of non-β-cell genes, including other islet hormones. This is partly due to altered cell composition of the islets, and partly due to altered gene signatures of the β-cells. An interactome screen identifies PAX6 as the TBL/R1 interaction partner responsible for regulating insulin gene expression. In humans, the islet gene expression of *TBL1X* correlates with HbA1c levels, and *TBL1X* and *TBL1XR1* gene variants are associated with high HbA1c across the population. The functional gene network around PAX6-TBL/R1 is conserved in human β-cells. Thus, we identify TBL/R1 as a regulatory component of β-cell identity and functionality necessary to maintain normoglycemia. Our findings elucidate the missing link between transcription factor activity and regulation of β-cell identity, with implications for human diabetes.

## Results

### Mice deficient in β-cell TBL/R1 display a diabetic phenotype

To determine the β-cell-specific function of the transcription co-factors TBL1X and TBL1XR1, we generated Ins1Cre TBL1X^fl/fl^ TBL1XR1^fl/fl^ mice (hereafter referred to as TBL/RβKO, Fig. 1a). Cre was inserted into the endogenous genomic *Ins1* locus[25], producing mice in which the Cre expression was restricted to the β-cells. Cre negative littermates were

used as controls. Ins1Cre expression alone did not affect β-cell function, insulin levels, or glucose metabolism, compared to Cre-negative littermates (Supplementary Fig. S1a–d). Reduced *Tbl1x* and *Tbl1xr1* gene expression was evident in isolated islets of TBL/RβKO mice at 5 weeks of age (Fig. 1b), but not in other tissues (Supplementary Fig. S1e).

Starting at 6-7 weeks of age, TBL/RβKO mice developed progressive hyperglycemia reaching a peak of above 500 mg/dL glucose at the age of 11 weeks (Fig. 1c). In line with an overt diabetes phenotype, TBL/RβKO mice showed reduced body weight due to reduced fat and lean mass and an activation of the catabolic program in liver and gonadal white adipose tissue (Supplementary Fig. S1f–i). The absence of insulin resistance (Supplementary Fig. S1j) suggested that defective insulin biosynthesis and/or secretion was causal for the hyperglycemia in TBL/RβKO mice. Indeed, in line with significantly elevated blood glucose levels in both fasted and refed TBL/RβKO mice (Fig. 1d), circulating insulin levels were lower in TBL/RβKO mice in comparison to control littermates, and remained unchanged upon refeeding (Fig. 1e). Total insulin levels in the pancreas were also significantly reduced in TBL/RβKO mice compared to control littermates (Fig. 1f). Refed circulating glucagon levels and total pancreas glucagon content were increased upon β-cell TBL/R1 knockout (Supplementary Fig. S1k, l).

At 5 weeks of age, when TBL/RβKO mice were still normoglycemic, they displayed a normal glucose tolerance and insulin secretion (Fig. 1g–i). In contrast, even at that age, TBL/RβKO mice displayed a high α/β-cell ratio with β-cell mass being significantly reduced while α-cell mass was unchanged (Fig. 1j and Supplementary Fig. S1m). Moreover, TBL/RβKO mice showed a disturbed islet morphology with α-cells dispersed throughout the islet rather than in the periphery (Fig. 1k). Thus, the islet morphology of TBL/RβKO mice resembles diabetic mouse islets even before the onset of hyperglycemia, indicating that TBL/R1 lack of function in β-cells may be causal for diabetes development.

At 13 weeks of age, when TBL/RβKO mice were hyperglycemic (Fig. 1c), they displayed poor glucose tolerance due to impaired insulin secretion (Fig. 1l, m). At 20 weeks of age, when hyperglycemia had persisted over a long period of time (Fig. 1c, g), the α/β-cell ratio was significantly elevated in islets of TBL/RβKO compared to control mice (Fig. 1n), and α-cells were dispersed throughout the islets rather than concentrated at the periphery, as in control islets (Fig. 1o). Also, at 20 weeks of age, α-cell mass was increased while β-cell mass was reduced (Supplementary Fig. S1n). Female mice at 20 weeks of age showed elevated circulating blood glucose levels, reduced β-cell mass, increased α/β-cell ratio with unchanged α-cell mass, and disturbed islet architecture, similar to male mice (Supplementary Fig. S2a–d). Of note, β-cell proliferation and apoptosis were unaffected in TBL/RβKO mice as assessed by BrdU and TUNEL staining, respectively (Supplementary Fig. S2e, f).

### TBL/R1 knockout in β-cells changes identity gene expression

To understand the underlying cause of the hyperglycemia and insulin depletion upon β-cell TBL1X and TBL1XR1 deficiency, we performed a transcriptomic analysis using islets from 5-week-old normoglycemic TBL/RβKO mice and control littermates. Although blood glucose levels were indistinguishable between the two groups (Supplementary Fig. S2g), 5860 genes were differentially expressed between control and TBL/RβKO islets, with 3209 upregulated and 2651 downregulated genes (Fig. 2a).

According to KEGG pathway analysis, differentially expressed genes were most significantly positively associated with an upregulation of the extracellular matrix (ECM)-receptor interaction (Fig. 2b and Supplementary Fig. S2h), previously associated with the loss of β-cell identity upon T2DM[26]. Further, KEGG pathways associated with glucose metabolism, such as maturity onset diabetes of the young (MODY), insulin secretion or type 2 diabetes mellitus, were significantly affected in TBL/RβKO islets (Fig. 2b and Supplementary Fig. S2i, j).

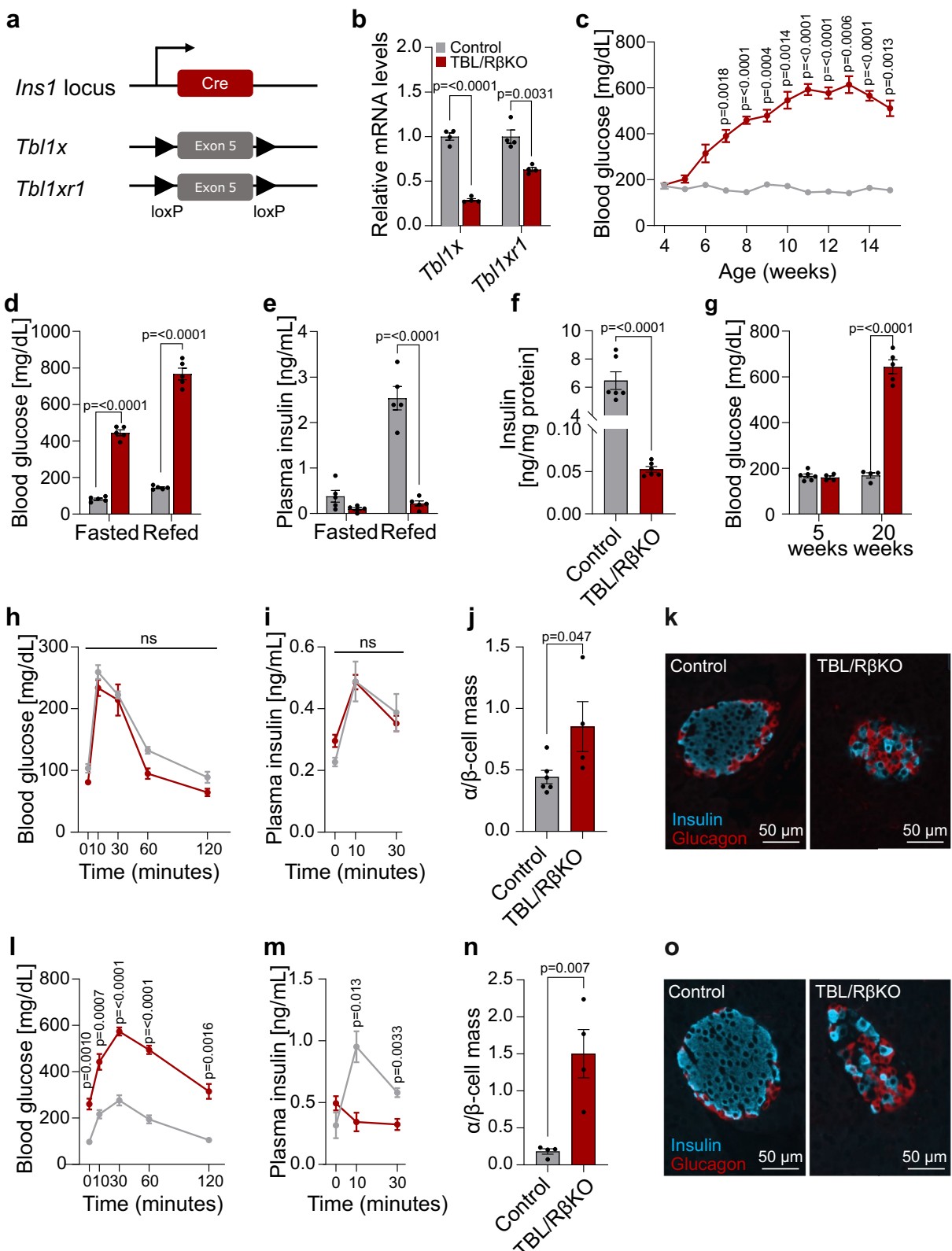

When focusing on genes with specific importance for pancreatic islets[10,13,27,28], we indeed observed a downregulation of the insulin and β-cell identity genes (e.g., *Mafa*, *Nkx6.1*, *Slc2a2*), while disallowed genes, progenitor genes, and genes associated with other cell types of the islets (e.g., *Ldha*, *Ngn3*, *ChgA*, *Gcg*, *Ppy*) were upregulated (Fig. 2c). This can either mean gene expression is changed within the β-cells or cell type composition of the islets is changed (i.e., less β-cells and more

non-β-cells). Gene set enrichment analysis demonstrated a significant ($q < 0.0001$, normalized enrichment score NES = 2.26) enrichment of genes associated with epithelial-mesenchymal transition – previously associated with dedifferentiated β-cells[26] - in TBL/RβKO islets, alongside a negative enrichment ($q < 0.0001$, NES = −2.06) of β-cell genes (Fig. 2d, e). This confirms our results from immunohistochemistry and overall suggests that TBL1X and TBL1XR1 are required for the

**Fig. 1 | Mice lacking β-cell TBL/R1 display a diabetes-like phenotype. a** Scheme for the generation of a β-cell-specific TBL1X and TBL1XR1 knockout. **b** Relative mRNA expression of *Tbl1x* and *Tbl1xr1* in pancreatic islets of TBL/RβKO (*n* = 4) and control mice (*n* = 4) at 5 weeks of age determined by qPCR. **c** Random fed blood glucose levels over time of TBL/RβKO (*n* = 6) and control mice (*n* = 6). **d, e** 16 h fasting and 2 h refeeding blood glucose (**d**) and plasma insulin (**e**) levels of TBL/RβKO (*n* = 5) and control (*n* = 5) mice at 15 weeks of age. **f** Total pancreas insulin content in TBL/RβKO (*n* = 6) and control (*n* = 6) mice at 15 weeks of age. **g** Random fed blood glucose levels of TBL/RβKO and control mice at 5 and 20 weeks of age. 5 weeks of age control *n* = 6, 5 weeks of age TBL/RβKO *n* = 4, 20 weeks of age control *n* = 4, 20 weeks of age TBL/RβKO *n* = 5. **h, l** Oral glucose tolerance test in TBL/RβKO (*n* = 8) and control (*n* = 5) mice at 5 (**h**) and 13 weeks (**l**) of age after 16 h of fasting. **i, m**, Plasma insulin levels determined during an oral glucose tolerance test in TBL/RβKO (*n* = 8) and control (*n* = 4 for 5 weeks and 5 for 13 weeks) mice at 5 (**i**) and 13 weeks (**m**) of age. **j, n**, α/β-cell mass ratio of pancreatic islets from TBL/RβKO and control mice at 5 (**j**) and 20 weeks (**n**) of age. 5 weeks of age control *n* = 6, 5 weeks of age TBL/RβKO *n* = 4, 20 weeks of age control *n* = 4, 20 weeks of age TBL/RβKO *n* = 4. **k, o** Representative immunofluorescent staining of insulin⁺ (blue, β-cells) and glucagon⁺ (red, α-cells) cells of paraffin-embedded pancreas from TBL/RβKO and control mice at 5 (**k**) and 20 weeks (**o**) of age. Data are represented as mean ± SEM. The following statistical tests were applied: two-sided student's *t* test (**b**, **f**, **g**, **j**, **n**), 2-way ANOVA with uncorrected Fisher's LSD *post hoc* test (**d**, **e**), and 2-way ANOVA with Šidák's multiple comparison *post hoc* test (**c**, **h**, **i**, **l**, **m**). ns – no significance. Source data are provided as a Source Data file.

regulation of transcriptional events determining the cellular identity of pancreatic β-cells.

To account for the multicellular nature of pancreatic islets, we next employed single-cell RNA-sequencing (scRNA-seq) of islets isolated from 5-week-old normoglycemic TBL/RβKO *vs.* control mice. UMAP based dimensionality reduction obtained a data structure showing four clearly separated islet cell populations enriched in insulin, glucagon, somatostatin and pancreatic polypeptide y transcripts, indicative of β-, α-, δ- and PP- cells, respectively, in both TBL/RβKO and control mice, as well as non-endocrine and mixed hormone expressing cell clusters (Fig. 3a). Transcriptional profiles of these four main populations are shown in Supplementary Fig. S3a–c.

Quantification of the identified endocrine cell populations revealed a reduction in β-cells and enrichment in α-, δ- and PP cells in TBL/RβKO islets (Fig. 3b, c), in line with the increased α/β-cell mass in TBL/RβKO islets observed by immunofluorescent staining (Fig. 1). Islets also contain a low percentage of polyhormonal cells, indicative of the cells' high plasticity and ability to de- /trans- differentiate across pancreatic endocrine cell fates[29]. This was previously described for diabetic islets[4]. We noticed an enrichment of cells positive for two or more islet hormones in the TBL/RβKO group, e.g., *Ins1/Gcg⁺*, *Ins1/Ppy⁺*, *Ins1/Sst/Ppy⁺* cells (Supplementary Fig. S3d), and confirmed the existence of *Ins1/Gcg⁺* cells in TBL/RβKO islets by immunofluorescent staining (Supplementary Fig. S3e).

When focusing on the β-cell population, pathway analysis of significantly altered genes ($\log_2$FC > 0.5, p-adj < 0.05) within the *Ins1⁺* cluster showed an enrichment of genes related to "gene expression in β-cells" and "β-cell development" in the downregulated genes (e.g., *Pdx1, Nkx6.1, Mafa, Slc2a2*), and an enrichment of genes related to stress signaling in the upregulated genes (e.g., *Jun, Me1*) (Supplementary Fig. S3f). UMAP based dimensionality reduction showed 6 β-cell clusters, and TBL/RβKO islets had reduced numbers of almost all β-cell clusters, while we identified one cluster that was virtually exclusive to TBL/RβKO cells (Fig. 3d, cluster 4, violet). This determined cluster was characterized by a low gene expression of the β-cell transcription factor *Mafa*, the glucose transporter *Slc2a2*, and the insulin genes, and a high expression of islet genes associated with diabetes, such as galectin-3-binding protein (*Lgals3bp*)[30] and glutamate-ammonia ligase (*Glul*)[31] (Fig. 3e and Supplementary Fig. S3h).

Aggregation of previously established marker gene expression within key islet processes into module scores[32] (Supplementary Data 1) identified immaturity as a pathway particularly enriched in the TBL/RβKO β-cell population, while maturity and immune attack susceptibility pathways were particularly diminished (Fig. 3f). The co-occurrence of the latter two pathways can be explained by a high semantic similarity (sim = 0.735) within the network and a high overlap in the genes defining the maturity and immune attack networks, with key genes being *Ins1, Ins2, Mafa, Nkx6.1*, and *Slc2a2*[32] (Supplementary Data 1).

Visualization of the maturity scores with ridgeline plots based on the β-cell clusters of TBL/RβKO and control islets further demonstrated the strong depletion of maturity characteristics in TBL/RβKO β-

cells (Fig. 3g), whereas immaturity characteristics were enriched (Supplementary Fig. S4a). Of note, we did not observe any differences in apoptosis signatures in TBL/RβKO *vs.* control β-cells (Supplementary Fig. S4b), in line with our previous results from immunofluorescent staining (Supplementary Fig. S2f).

Gene set enrichment analysis based on GO terms showed that among the most significantly regulated processes in TBL/RβKO β-cells were extracellular matrix binding and ion channel activity (Supplementary Fig. S4c), highlighting that the regulated pathways identified in the bulk RNA-seq (Fig. 2) were primarily due to regulation in β-cells. Deconvolution of bulk sequencing data further confirmed a change in both cell type composition and β-cell intrinsic gene expression, with "mature" and "extreme" β-cell phenotypes most significantly down-regulated (Supplementary Fig. S4d, e). These data overall support the notion that β-cells lacking TBL1X and TBL1XR1 lose their cellular identity, reverting to a more undifferentiated state or alternative cell fate with an expression profile of less mature β-cells. Thus, alterations in cellular identity upon TBL/R1 loss of function is the likely cause for the reduced β-cell numbers and progressive hyperglycemia observed in TBL/RβKO mice.

## TBL/R1 loss in adult β-cells produces progressive diabetes

Pancreatic β-cells undergo several steps of maturation to acquire their final identity, beginning in embryonic stages[33,34] and finalizing around the time of weaning triggered by the transition from high fat mother's milk to high carbohydrate chow diet[35]. As TBL/RβKO mice showed a disturbed islet architecture compared to control mice at 5 weeks of age (shortly after weaning, Fig. 1) before developing hyperglycemia at 6-7 weeks of age, we investigated whether the discrete maturation step associated with the dietary change at weaning affected the TBL/R1-dependent islet morphology. Neither premature weaning nor extension of the suckling period by 7 days altered the islet phenotype previously observed in TBL/RβKO mice (Supplementary Fig. S4f, g and Fig. 1j, k), suggesting that TBL/R1 regulates β-cell identity independent of weaning.

Therefore, to explore whether TBL1X and TBL1XR1 are important for the maintenance of β-cell identity also in adult mice, we produced mice carrying a tamoxifen-inducible β-cell-specific knockout of TBL1X and TBL1XR1 (iTBL/RβKO). Cre expression was under the rat *Ins2* promoter (*Rip2*) (Fig. 4a)[36]. When induced in young adult mice (8 weeks) and challenged with a 60% high fat diet (HFD), iTBL/RβKO produced progressive hyperglycemia (Fig. 4b) with unchanged body weight and body composition, and slightly improved insulin tolerance (Supplementary Fig. S5a–c), compared to HFD-fed control mice.

Although the iTBL/RβKO mice maintained normoglycemia for weeks after induction of TBL/R1 knockout, pancreatic insulin content was reduced already 4 weeks after induction, and this was aggravated at 11 weeks post induction (Fig. 4c). Glucose tolerance was reduced 22 weeks post induction (Fig. 4d) due to impaired insulin secretion (Fig. 4e). Reduced expression of *Tbl1x* and *Tbl1xr1* genes was confirmed in pancreatic islets but not any other tissues (Fig. 4f, Supplementary Fig. S5d, e).

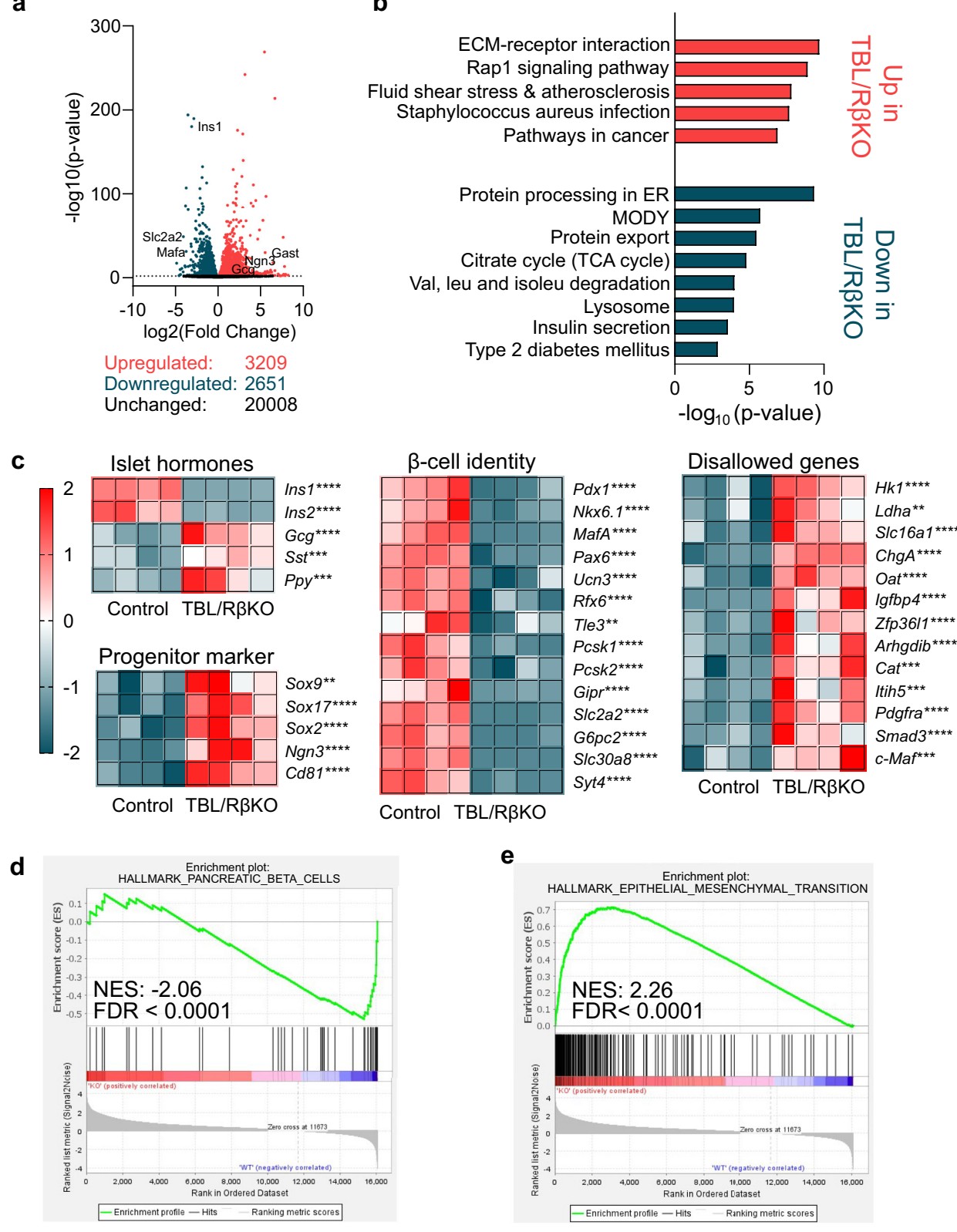

**Fig. 2 | Pancreatic islets of TBL/RβKO mice show changes in transcriptional events associated with loss of β-cell identity and function. a** Volcano plot displaying differentially expressed genes from TBL/RβKO and control islets isolated from 5-week-old normoglycemic mice from bulk RNA sequencing. **b** Top up- and downregulated pathways identified by KEGG pathway analysis. **c** Heatmap of differentially regulated genes of islet hormones, progenitor markers, β-cell identity genes, and disallowed genes in islets of TBL/RβKO and control mice. Each box represents one mouse. Color indicates a z-score for TBL/RβKO vs. control islets. **d**, **e** Gene set enrichment analysis indicates a downregulation of genes associated with β-cell function (**d**) and an upregulation of genes associated with the epithelial mesenchymal transition (**e**) in islets of TBL/RβKO mice. p-values were adjusted applying the Benjamini-Hochberg's method for controlling the false discovery rate (FDR). **p < 0.01, ***p < 0.001, ****p < 0.0001.

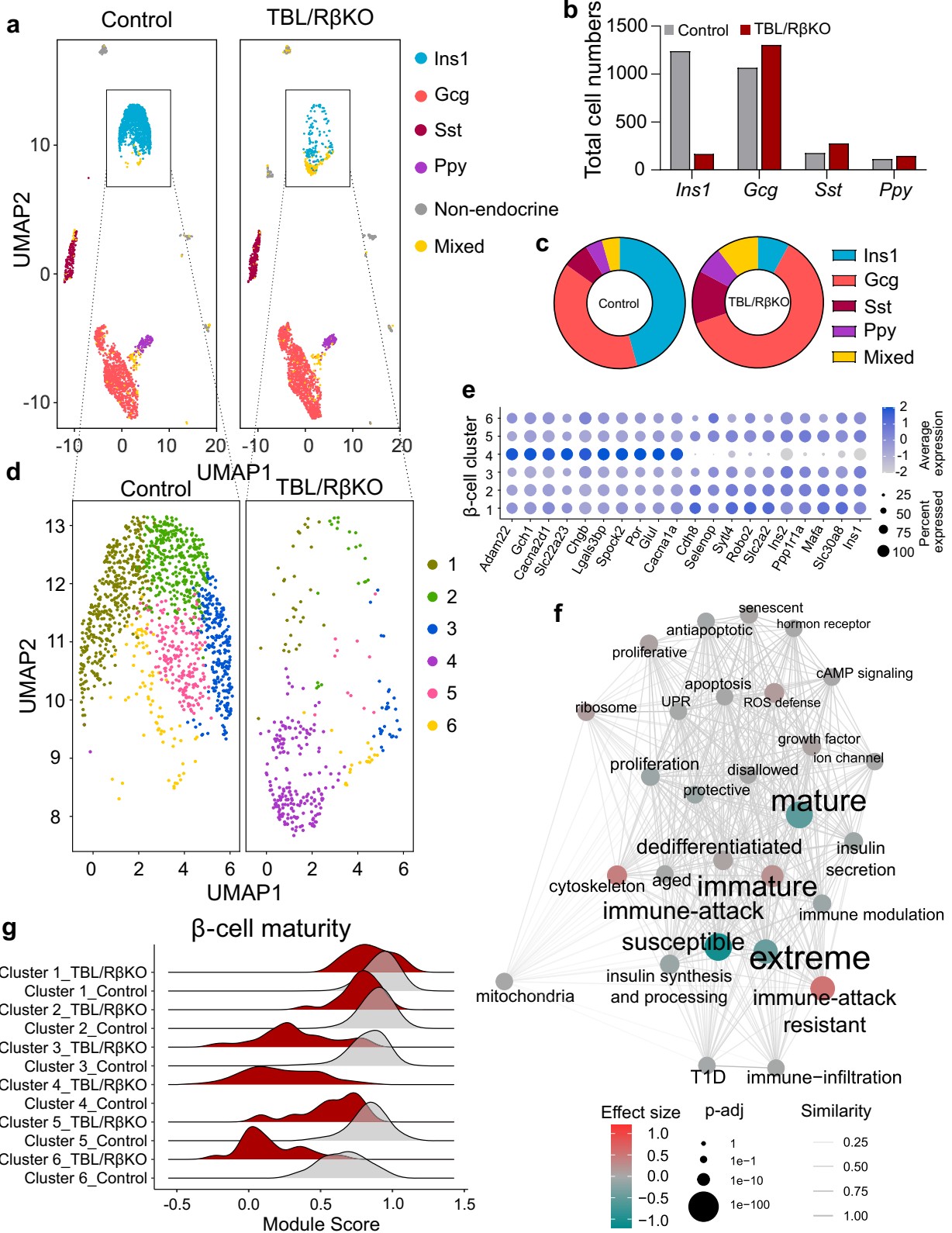

Expression of genes determining β-cell identity (e.g., *Ins1, Ins2, Nkx6.1, Slc2a2, Mafa, Pdx1*) were significantly reduced in islets of iTBL/RβKO mice, whereas genes not typically expressed in β-cells (e.g., *Ldha, Hk1*), were induced (Fig. 4g). Overall, this resembled the gene expression profile in islets of TBL/RβKO mice. Likewise, the islet morphology of iTBL/RβKO resembled that of TBL/RβKO mice,

with an increased α/β-cell ratio and α-cells dispersed throughout the islet (Fig. 4h, i). In summary, induction of TBL/R1 loss of function in adult β-cells produced a phenotype similar to the non-inducible loss of function but with altered dynamics, highlighting that the role of TBL/R1 in maintaining β-cell identity is developmental stage and model (i.e., promoter) independent.

**Fig. 3 | β-cells deficient for TBL/R1 are distinct in their transcriptional signature in comparison to control β-cells. a** UMAP plot showing data structure of single-cell RNA-sequencing (scRNA-seq) data of islets from normoglycemic TBL/RβKO and control mice (pooled from 4 mice/group) at the age of 5 weeks. Endocrine cell populations are labeled based on islet cell identity using known marker genes (glucagon (*Gcg*), insulin (*Ins1*), pancreatic polypeptide y (*Ppy*), somatostatin (*Sst*), multiple markers (Mixed)). Non-endocrine cells were summarized in one cell population. Each dot represents one cell. **b** Bar graph showing the total number of the 4 major islet cell types in TBL/RβKO and control mice identified by scRNA-seq. **c** Pie chart showing the relative abundance of the 4 major islet cell types and polyhormonal cells (Mixed) in TBL/RβKO and control mice identified by scRNA-seq.

**d** UMAP plot showing clustering within the β-cell population highlighted in (**a**), which resulted in 6 distinct clusters. **e** Most significantly up- and downregulated genes in cluster 4 in comparison to the other β-cell clusters. Color indicates average expression, dot size indicates the percentage of cells found to express the indicated gene. **f** Semantic similarity network of β-cell heterogeneity terms with statistical analysis of aggregated gene expression into module scores between β-cells from TBL/RβKO and control mice. Edge weights indicate similarities between terms, color indicates effect size and direction, and dot size indicates significance between the experimental groups. **g** Ridgeline plot showing the module score of genes associated with β-cell maturity within the different clusters identified in (**c**) in control and TBL/RβKO mice.

## TBL1X and TBL1XR1 form a gene regulatory network with PAX6

Being transcription co-factors, TBL/R1 interact with a broad range of nuclear receptors and transcription factors in a tissue-specific manner[17,23,37,38] to fine-tune their transcriptional activity. Thus, to understand how TBL/R1 regulate the expression of β-cell identity genes, we performed an interactome analysis of both proteins in the murine β-cell line MIN6.

Using endogenous immunoprecipitation followed by liquid chromatography with tandem mass spectrometry (LC-MS/MS), we found that nuclear proteins interacting with TBL/R1 were associated with transcriptional regulation, pattern specification and pancreas development, in line with our previous results (Fig. 5a, b). Verifying the quality of our interactome screen, we identified several previously established components of the NCOR/SMRT complex[19,20,37] as TBL1X and TBL1XR1 interaction partners, such as HDAC3, NCOR1, NCOR2, and GPS2 (Tables 1, 2). Further, we pulled down components of the facilitates chromatin transcription (FACT) transcriptional activation complex, including SPT16, indicating that TBL/R1 can be involved in both transcriptional repression and activation as previously noted[18].

A transcription factor of particular interest that we found to interact with both TBL1X and TBL1XR1 was paired box 6 (PAX6) (Tables 1, 2), an established master-regulator of islet gene expression[9,39] not previously associated with TBL/R1. We confirmed the interaction of PAX6 with TBL1X and TBL1XR1 in MIN6 cells using immunoprecipitation followed by western blot (Supplementary Fig. S6a). Further, we confirmed the interaction of PAX6 with TBL1X, TBL1XR1, HDAC3 and SPT16 in the rat insulinoma cell line INS1E, using immunoprecipitation followed by western blot (Fig. 5c).

## TBL/R1 are required for PAX6 mediated insulin gene expression

Considering the strong repression of insulin gene expression in TBL/RβKO and iTBL/RβKO mice, we hypothesized that TBL/R1 directly regulates PAX6-dependent insulin promoter activity. Indeed, siRNA-mediated TBL/R1 knockdown (Fig. 5d) reduced murine *Ins2* (*mIns2*) promoter activity (Fig. 5e). The overexpression of PAX6 increased *mIns2* promoter activity, in line with the previously reported transcriptional activation of insulin gene expression by PAX6[9,39]. Chromatin immunoprecipitation showed that both TBL/R1 and PAX6 bound directly to the *mIns2* promoter in INS1E cells (Supplementary Fig. S6b). TBL/R1 knockdown completely abolished the PAX6-induced *mIns2* promoter activity, highlighting that TBL/R1 are required for PAX6-mediated transcriptional activation of the insulin gene in β-cells (Fig. 5e). In contrast, we observed no direct regulation of the murine *Mafa* promoter by TBL/R1, irrespective of PAX6 expression (Supplementary Fig. S6c).

Next, to establish a link between PAX6, TBL/R1, and the identified transcriptional repressor/ activator complexes, we assessed PAX6 binding to HDAC3 and SPT16 under TBL/R1 knockdown. Indeed, PAX6 interacted with both HDAC3 and SPT16, and this was dependent on TBL/R1, as TBL/R1 knockdown increased PAX6 binding to HDAC3 and reduced PAX6 binding to SPT16 (Fig. 5f).

The HDAC3 specific inhibitor (HDAC3i) BRD3308[40] reduced *mIns2* promoter activity in non-transfected and PAX6 overexpressing, but not TBL/R1 knockdown cells (Fig. 5g). This was confirmed by the alternative HDAC3 inhibitor RGFP966 (Supplementary Fig. S6d). These data indicate that HDAC3 mediates TBL/R1-dependent activation of insulin gene transcription via PAX6. Indeed, TBL/R1 knockdown caused reduced *Ins2* gene expression in INS1E cells (Fig. 5h). In contrast, overexpression of TBL1XR1 in INS1E cells caused a significant increase in glucose-stimulated insulin secretion (Fig. 5i, Supplementary Fig. 6e), which was conserved in TBL1XR1 overexpressing primary murine islets, alongside a mild, non-significant increase in *Ins2* gene expression (Fig. 5j, k, S6i). Of note, TBL1X overexpression in INS1E and EndoC-βH1 cells only mildly affected basal insulin secretion (Supplementary Fig. S6e–i).

## TBL/R1 regulate human insulin gene transcription via PAX6

Data thus far stem from mouse models and rodent β-cells, hence we sought to verify the identified mechanism in human β-cells. As in rodent cells, TBL1X, TBL1XR1 and PAX6 bound to the human insulin promoter (*hINS*) in human EndoC-βH1 cells (Fig. 6a). The weaker signal for TBL1X and TBL1XR1 in comparison to PAX6 may be due to indirect binding of TBL/R1 to the DNA as part of a regulatory complex, as is typical for transcription co-factors. In the same cells, PAX6 interacted with TBL1XR1, HDAC3 and SPT16, albeit not TBL1X (Fig. 6b). TBL/R1 knockdown reduced human insulin promoter activity (Fig. 6c). Lastly, TBL/R1 knockdown also reduced the insulin secretion of EndoC-βH1 cells (Fig. 6d). These data demonstrate that the TBL/R1-mediated regulation of insulin levels via PAX6 is conserved between rodents and humans.

## Human *TBL1X* variants and gene expression associate with HbA1c

Impaired insulin expression can cause hyperglycemia and diabetes development in patients[41]. Thus, we hypothesized that alterations in TBL/R1 gene expression or activity may contribute to diabetes development. Indeed, in human organ donors, we observed a negative correlation between glycated hemoglobin A1c (HbA1c) levels and the *TBL1X* gene expression in isolated islets (Fig. 6e). In line with this, islets of diabetic db/db mice or aged animals - which are prone to develop glucose intolerance[42] also expressed lower *Tbl1x* mRNA levels (Supplementary Fig. S7a, b). The correlation between HbA1c and islet *TBL1XR1* gene expression in human islets was not significant (Supplementary Fig. S7c).

Single-nucleotide polymorphisms in the genetic region of both *TBL1X* and *TBL1XR1* genes were associated with elevated HbA1c levels (Fig. 6f, g). Further, variants near the *TBL1X* and *TBL1XR1* genes were associated with random blood glucose levels (Supplementary Fig. S7d, e). Overall, our data establish TBL/R1 as regulators of critical β-cell genes important for the maintenance of β-cell identity and normoglycemia and demonstrate that alterations in TBL/R1 levels or function contribute to diabetes development.

## Discussion

How β-cell transcription factors function simultaneously as activators and repressors within the same cell is an important open

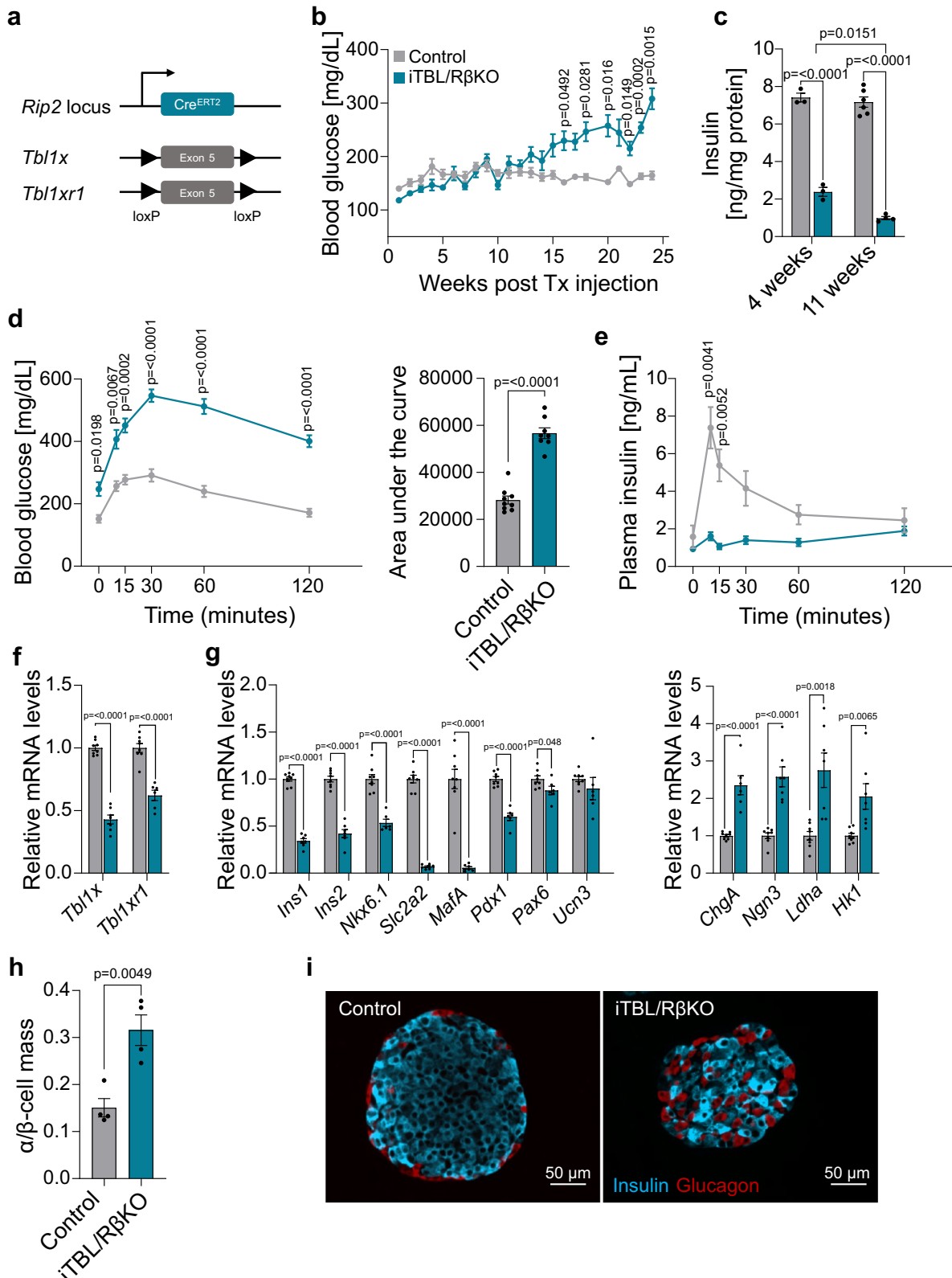

question with wider implications for diabetes development and progression, as these transcription factors determine β-cell function and ultimately adequate insulin secretion. The key to understanding these regulatory processes lies in understanding the functional gene networks that modulate β-cell transcription factor activity.

In the present study, we identified TBL/R1 as transcriptional regulators important for pancreatic β-cell identity with relevance for human T2DM. β-cell-specific knockout of TBL/R1 in two independent mouse models caused progressive hyperglycemia and hypoinsulinemia due to impaired expression of critical β-cell identity genes. The presence of TBL/R1 was required to maintain functional β-cell mass.

**Fig. 4 | Induction of TBL/R1 loss of function in adult β-cells produces progressive diabetes due to impaired maintenance of β-cell identity. a** Scheme for the generation of an inducible β-cell-specific TBL1X and TBL1XR1 knockout. **b** Random fed blood glucose levels over time in iTBL/RβKO ($n = 9$) and control ($n = 9$) mice on HFD. **c** Total pancreas insulin content normalized to protein levels in iTBL/RβKO and control mice 4 and 11 weeks after tamoxifen administration. $n = 3$ for 4w control and iTBL/RβKO mice, $n = 6$ for 11w control mice, $n = 4$ for 11w iTBL/RβKO mice. **d, e** Blood glucose (**d**) and plasma insulin (**e**) levels during an oral glucose tolerance test after 22 weeks of HFD. Corresponding area under the curve in iTBL/RβKO ($n = 8$) and control ($n = 9$) mice (**d**, right). **f, g** Relative mRNA expression determined by qPCR in pancreatic islets of iTBL/RβKO and control mice of *Tbl1x* and *Tbl1xr1* (**f**) and islet genes (**g**). Control mice *Tbl1x, Tbl1xr1, Ins1, Ins2,*

*Nkx6.1, Slc2a2, Mafa, Pdx1, Pax6, Ucn3*: $n = 9$, control mice *ChgA, Ngn3, Ldha, Hk1*: $n = 8$, iTBL/RβKO mice *Tbl1x, Ngn3, Ldha, Hk1*: $n = 7$, iTBL/RβKO mice *Tbl1xr1, Ins1, Ins2, Slc2a2, Mafa* $n = 6$, iTBL/RβKO mice *Nkx6.1, Pdx1, Pax6, Ucn3, ChgA*: $n = 5$. **h** α/β-cell mass ratio of pancreatic islets from iTBL/RβKO ($n = 4$) and control mice ($n = 4$) on HFD, 24 weeks after knockout induction. **i** Representative immuno-fluorescent staining of insulin⁺ (blue, β-cells) and glucagon⁺ (red, α-cells) cells of paraffin-embedded pancreas from iTBL/RβKO and control mice on HFD, 24 weeks after knockout induction. Each point represents one mouse. Data are represented as mean ± SEM. The following statistical tests were applied: two-sided student's *t* test (**d** – Area under the curve, **f–h**) and 2-way ANOVA with Šidák's multiple comparison *post hoc* test (**b–d** – time course, **e**). Source data are provided as a Source Data file.

---

We found that TBL/R1 exerted their function through modulating PAX6 transcriptional activity in both rodent and human β-cells, in concert with the transcription activators and repressors FACT and NCOR/SMRT, respectively.

Animals with β-cell TBL/R1 deficiency developed hyperglycemia due to reduced insulin production and secretion, despite normal initial glucose or insulin tolerance. This was marked by low pancreas insulin content and low secretion during feeding or glucose tolerance tests. In addition, glucagon levels were higher, and the fasting-feeding regulation was disturbed. The hyperglycemia in TBL/RβKO mice was preceded by strong and significant alterations in islet and β-cell gene expression. This is relevant as even small changes in glucose concentrations can have profound transcriptional consequences including on β-cell identity genes[43,44]. Gene signatures of TBL/RβKO islets resemble those of diabetic islets, with an enrichment of pathways, hallmarks, and genes typical for diabetic islets[12,32,43]. Furthermore, before the onset of hyperglycemia, TBL/RβKO mice had an increased α/β-cell ratio and disrupted islet morphology, featuring a lower β-cell mass and scattered α-cells, similar to diabetic animals[5,45]. TBL/RβKO mice therefore have islets with typical features of diabetic mice, but before any phenotypic changes typical for diabetes development are apparent (hyperglycemia, impaired glucose tolerance).

Single-cell RNA-sequencing in mice lacking TBL/R1 in pancreatic β-cells showed the disappearance of mature β-cells and the emergence of immature and polyhormonal cells. These likely de- or transdifferentiated cells have been previously observed[29], particularly during diabetes progression, where trajectory analysis of endocrine cells indicated a shift in β-cell identity through dedifferentiation and transdifferentiation[46]. A similar process is suggested in early-stage TBL/RβKO islets, marked by their immature gene expression profiles. It should be noted, however, that the total numbers of these polyhormonal cells are low, and we cannot exclude that this population also contains cell doublets despite computational doublet removal. Deconvolution of bulk RNA-sequencing data should be interpreted with caution due to inherent methodological limitations; yet bulk-sequencing confirmed reduced "maturity" signatures in TBL/RβKO β-cells. Beginning in embryonic stages, pancreatic β-cells undergo several steps of differentiation and maturation to acquire their final identity[33–35,47]. Cre is expressed under the endogenous *Ins1* promoter, hence expression is initiated at E12.5[33] when insulin starts being transcribed, i.e., when the cells are differentiated into β-cells. Our model therefore does not provide insights into TBL/R1's role in earlier differentiation steps, but places TBL/R1 as regulators of β-cell identity during weaning-independent maturation.

In TBL/RβKO mice, we have not resolved if β-cells lose their identity, or if they fail to develop their identity. However, iTBL/RβKO mice, in which β-cell-specific TBL/R1 knockout is induced in adulthood, show similar changes in islet gene expression and morphology as TBL/RβKO mice. This implies that TBL/R1 are important for maintaining functional β-cells, rather than (or in addition to) affecting the initial maturation of these cells. In this respect, it is worth highlighting that apoptosis was not different between TBL/RβKO and control islets and

β-cells, underlining that a loss of β-cell identity rather than loss of general cell mass is responsible for the progressive deterioration of functional β-cell mass leading to diabetes.

A prominent feature of TBL/RβKO islets was the reduced expression of the insulin genes, and through promoter binding and activity assays, we found a direct regulation of insulin promoters through TBL/R1, together with PAX6. PAX6 has previously been described as a developmental regulator essential for the maintenance of islet cell function in adult pancreas[9,48,49]. Indeed, the induction of a global PAX6 depletion in adult mice caused a very rapid-onset diabetes[49]. Later studies investigating PAX6 function specifically in β-cells showed that PAX6 is a transcriptional regulator of insulin gene expression[39], and directly bound to the insulin promoter[9,50]. The depletion of PAX6 from β-cells in adult mice caused progressive hyperglycemia and hypoinsulinemia[39], as well as a loss of β-cell function and expansion of α-cells and other islet cells[9]. This phenotype is remarkably similar to that of TBL/RβKO mice. While these previous studies demonstrated that PAX6 was required for the maintenance of the functional β-cell identity, they did not uncover the higher level gene regulatory network responsible for its function, and indeed posed the question how the PAX6 transcription factor can act as both activator and repressor of gene expression[9].

We here demonstrate that the transcription activator function of PAX6 on the insulin promoter is dependent on its interaction with the transcriptional co-factor TBL/R1. This seemed to be specific to the insulin promoter, as neither *Mafa* nor glucagon (data not shown) promoter activity were directly regulated by TBL/R1 – opposite to PAX6. This allows the following conclusions. A) changes to additional β-cell genes and disallowed genes in TBL/RβKO islets are likely secondary to alterations in insulin gene expression (although amplifying feedback loops may exist). B) TBL/R1 is a regulatory factor of β-cell function and identity relevant for diabetes development. C) the TBL/R1-PAX6 gene regulatory network is important for only a subset of PAX6 targets and thus our study identifies TBL/R1 as prototypic co-factors representing an additional layer of regulatory program in β-cells. This has broader implications for diabetes, as targeting established β-cell transcription factors has not yet resulted in therapeutic options for T2DM[51]. Perhaps the most promising recent development in diabetes research is the prospect of islet transplantation, with islets derived from human pluripotent stem cells (hPSCs)[51]. One remaining challenge here is their still inefficient and sometimes unreproducible differentiation, causing impaired physiologic functionality[52,53]. Exploring the direct targeting of a regulatory layer of β-cell identity in humans, or the involvement of TBL/R1 in proper hPSC differentiation, could thus present a promising avenue for treatment.

Our study identified further components of the PAX6 gene regulatory network: HDAC3 mediated TBL/R1-dependent activation of insulin gene expression via PAX6, as inhibition of HDAC3 reduced the insulin promoter activity, and this was dependent on the presence of TBL/R1 (Fig. 5). These data are in line with observations from β-cell-specific HDAC3 knockout mice that develop progressive diabetes

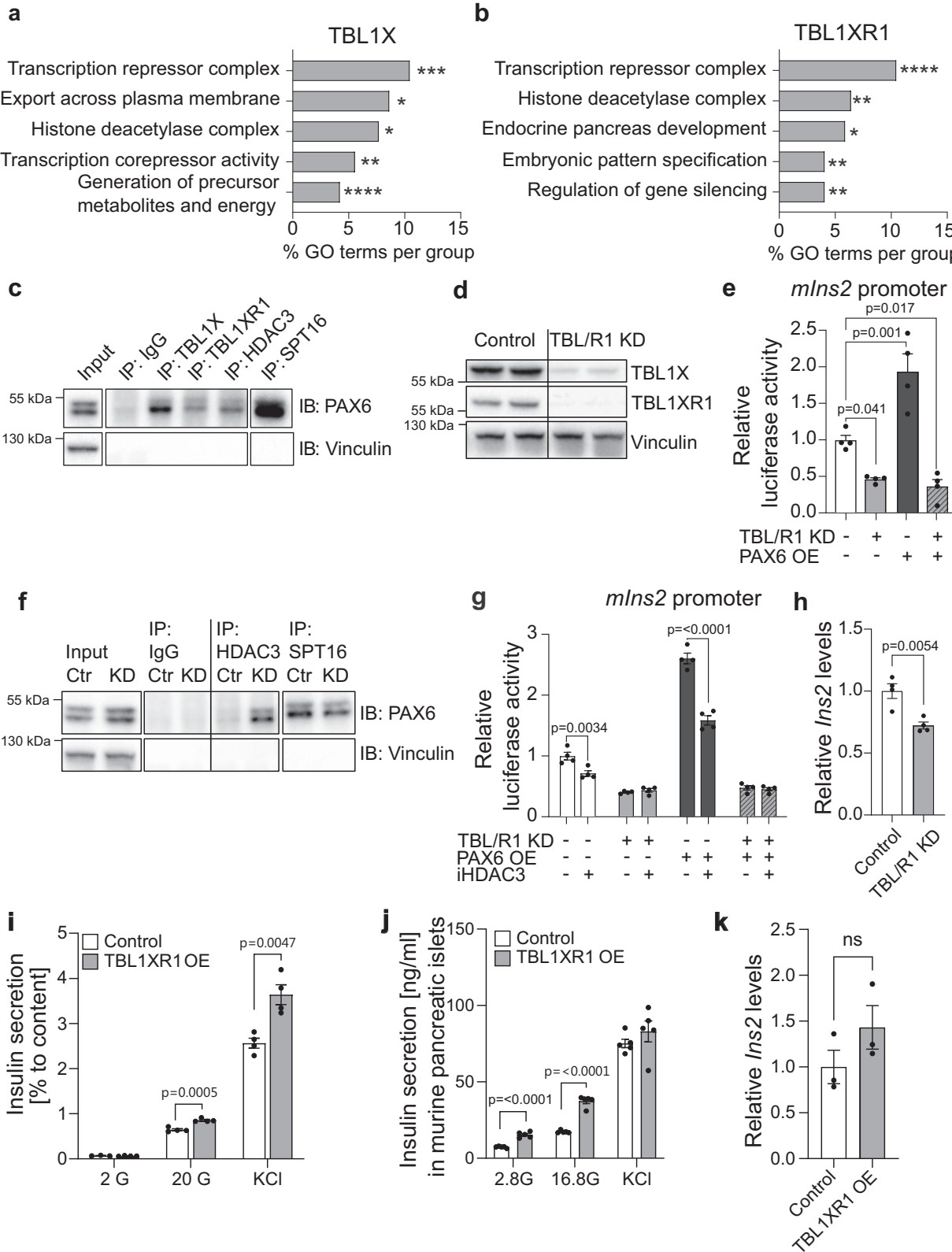

through unknown mechanisms[54]. HDAC3 is mainly described as core component of the NCOR/SMRT repressor complex[20] together with TBL/R1. Thus, the role of HDAC3 as well as TBL/R1 as transcriptional activators is somewhat unexpected. There have been rare reports of transcriptional activation by NCOR/SMRT components in the past[55], and these are in line with the direct insulin promoter repression by TBL/R1 and HDAC3 reduction described here.

We showed that the *TBL1X* gene expression in islets was lower with higher HbA1c, and low *Tbl1x* expression was also observed in islets of diabetic db/db animals (Supplementary Fig. S7a). Thus, a self-amplifying loop of elevated glycemia – low *TBL1X* expression is conceivable, eventually culminating in diabetes development. In line with this, *TBL/R1* gene variants are significantly associated with high HbA1c and blood glucose levels in humans. This overall indicates that TBL/R1

**Fig. 5 | TBL1X and TBL1XR1 are regulators of PAX6 mediated gene expression.**
**a**, **b** 5 selected metabolic processes identified via gene ontology analysis using significantly enriched TBL1X (**a**) and TBL1XR1 (**b**) interaction partners with a unique peptide count > 2 identified in the interactome screen. **c**, Endogenous TBL1X, TBL1XR1, HDAC3, and SPT16 was immunoprecipitated in INS1E cell lysates with rabbit IgG as negative control. Whole cell lysates (Input) and immunoprecipitated proteins were immunoblotted using the indicated antibodies. **d** TBL1X, TBL1XR1, and vinculin protein levels in INS1E cells after siRNA transfection. **e** Murine *Ins2* (*mIns2*) promoter activity in INS1E cells upon TBL/R1 knockdown or/and PAX6 overexpression. Luciferase activity normalized to renilla luciferase. *n* = 4 wells/condition. **f** Endogenous HDAC3 and SPT16 was immunoprecipitated in INS1E cell lysates upon TBL/R1 knockdown with rabbit IgG as a negative control. Whole cell lysates (Input) and immunoprecipitated proteins were immunoblotted using the indicated antibodies. **g** Murine *Ins2* (*mIns2*) promoter activity in INS1E cells upon TBL/R1 knockdown and/or PAX6 overexpression with/ without HDAC3 inhibitor (BRD3308, 1 μM, 24 h). Luciferase activity normalized to renilla luciferase. *n* = 4

wells/condition. **h** Relative *Ins2* gene expression in INS1E cells upon TBL/R1 knockdown determined by qPCR. *n* = 4 wells/condition. **i** Insulin secretion relative to insulin content in INS1E cells upon *Tbl1xr1* overexpression. Insulin secretion was stimulated using 2 mM glucose (2 G), 20 mM glucose (20 G), and 2 mM glucose supplemented with KCl (KCl). Control 2 G *n* = 3 wells/condition, all other data *n* = 4 wells/condition. **j** Insulin secretion in murine pancreatic islets upon *Tbl1xr1* over-expression. Insulin secretion was stimulated using 2.8 mM glucose (2.8 G), 16.8 mM glucose (16.8 G), and 2.8 mM glucose supplemented with KCl (KCl). *n* = 5. **k** *Ins2* gene expression determined by qPCR in murine pancreatic islets upon adenovirus-mediated *Tbl1xr1* overexpression. *n* = 3 wells/condition. Data are represented as mean ± SEM. The following statistical tests were applied: 1-way ANOVA with Dunnett's multiple comparison *post hoc* test (**e**), 2-way ANOVA with a Tukey's multiple comparison *post hoc* test (**g**), and a two-sided student's *t* test (**h–k**). ns = no significance,*p < 0.05 **p < 0.01, ***p < 0.001, ****p < 0.0001. Source data are provided as a Source Data file.

### Table 1 | Identified TBL1X interaction partners in pancreatic β-cells

| Protein name | Protein description | Unique peptide count | Ratio TBL1X/Flag | p-value |
|---|---|---|---|---|
| PWAR | PRKC apoptosis WT1 regulator protein | 4 | 231,0 | 0.00352 |
| NISCH | Nischarin | 9 | 39,0 | < 0.0001 |
| U5S1 | 116 kDa U5 small nuclear ribonucleoprotein component | 2 | 27,2 | < 0.0001 |
| HDAC3 | Histone deacetylase 3 | 14 | 26,0 | 0.0002 |
| NCOR2 | Nuclear receptor corepressor 2 | 44 | 21,3 | 0.00038 |
| GPS2 | G protein pathway suppressor 2 | 7 | 17,9 | 0.00018 |
| NCOR1 | Nuclear receptor corepressor 1 | 46 | 17,1 | 0.00101 |
| UBP28 | Ubiquitin carboxyl-terminal hydrolase 28 | 2 | 15,9 | < 0.0001 |
| UPP | Uracil phosphoribosyltransferase homolog | 3 | 14,3 | 0.00047 |
| PAX6 | Paired box protein 6 | 3 | 12,4 | 0.00547 |
| QKI | Protein quaking | 3 | 6,8 | < 0.0001 |
| KDMA1 | Lysine-specific histone demethylase 1 A | 6 | 6,1 | 0.00147 |
| RCOR3 | REST corepressor 3 | 2 | 5,2 | 0.00381 |
| UBE2N | Ubiquitin-conjugating enzyme E2 | 3 | 4,6 | 0.01266 |
| TBL1XR1 | F-box-like/WD repeat-containing protein TBL1XR1 | 12 | 4,6 | 0.00293 |
| MVP | Major vault protein | 4 | 3,9 | 0.02067 |
| IQGA1 | Ras GTPase-activating-like protein | 8 | 3,9 | 0.06367 |
| VAPA | Vesicle-associated membrane protein-associated protein | 2 | 3,9 | 0.00159 |
| PRP8 | Pre-mRNA-processing-splicing factor 8 | 2 | 3,4 | 0.04118 |
| H32 | Histone H3.2 | 2 | 3,3 | 0.00239 |

Top 20 nuclear TBL1X interaction partners. After TBL1X enrichment with an endogenous immunoprecipitation, interaction partners were identified using mass spectrometry. The ratio TBL1X/Flag shows the enrichment of the respective protein in the TBL1X pulldown relative to the enrichment in the pulldown using Flag antibody. Statistical analysis was performed using a two-sided student's *t* test. *n* = 4 wells per condition.

are specifically involved in glucose homeostasis, and altered function or expression of these genes may contribute to diabetes development and progression in patients. Of note, the genetic analysis does not allow to draw direct conclusions on pancreatic TBL/R1 function, and further, TBL1X and TBL1XR1 expression within the pancreas is not restricted to β-cells, but the co-factors are also expressed in other islet cell types in both mice (our scRNA-seq data) and humans[56]. The relative contribution of the different islet cell types for TBL/R1 gene expression (and its link to diabetes phenotypes) will be subject to future studies.

In line with the previously discovered function of the TBL/R1 co-factor complex in hepatic metabolism and adipose tissue lipid handling[17,23], and its role in cell fate decisions[57], our current findings overall position the TBL/R1 transcriptional co-factors as important components of the cellular identity program in pancreatic β-cells and critical node in systemic glucose homeostasis. Fine-tuning gene expression in a promoter-specific manner by co-factors such as TBL/R1 may therefore explain how β-cell transcription factors function

simultaneously as activators and repressors within the same cell and why targeting β-cell transcription factors has so far been ineffective as diabetes therapy in patients[58]. As loss of β-cell identity is a reversible process[14,59,60], a normalization of the diabetes phenotype by TBL/R1 regulation could be conceivable. The association of TBL/R1 variants with glycemia parameters in human subjects indicates that tissue-specific targeting of the TBL/R1 gene network or manipulating TBL/R1 during β-cell differentiation from hPSCs may represent a conserved, yet unappreciated, approach to counteract diabetes.

In conclusion, the current study identified TBL/R1 as regulators of β-cell gene expression relevant for the maintenance of functional β-cells in mice and humans. Future studies will aim at identifying approaches to target the TBL/R1 regulatory network to counteract diabetes.

## Methods
Our research complies with all relevant ethical regulations as outlined in the 'Animal experiments' and 'Pancreatic islet isolation' sections.

**Table 2 | Identified TBL1XR1 interaction partners in pancreatic β-cells**

| Protein name | Protein description | Unique peptide count | Ratio TBL1XR1/Flag | p-value |
|---|---|---|---|---|
| IF4E | Eukaryotic translation initiation factor 4E | 2 | 28,1 | <0.0001 |
| TBL1XR1 | F-box-like/WD repeat-containing protein | 12 | 11,4 | <0.0001 |
| PAX6 | Paired box protein 6 | 3 | 9,7 | 0.0088 |
| HDAC3 | Histone deacetylase 3 | 14 | 9,6 | 0.0004 |
| MEOX2 | Homeobox protein MOX-2 | 3 | 6,8 | 0.0045 |
| NCOR2 | Nuclear receptor corepressor 2 | 44 | 5,2 | 0.0002 |
| NCOR1 | Nuclear receptor corepressor 1 | 46 | 5,1 | 0.0016 |
| DLDH | Dihydrolipoyl dehydrogenase | 9 | 4,8 | 0.0022 |
| GPS2 | G protein pathway suppressor 2 | 7 | 4,7 | 0.0038 |
| OSB5 | Proteasome subunit beta type-5 | 2 | 3,3 | 0.0105 |
| CUX1 | Homeobox protein cut-like 1 | 3 | 3,2 | 0.0037 |
| UACA | Uveal autoantigen with coiled-coil domains and ankyrin repeats | 35 | 3,1 | 0.0004 |
| RMI1 | RecQ-mediated genome instability protein 1 | 3 | 2,9 | 0.0104 |
| RBBP4 | Histone-binding protein RBBP4 | 2 | 2,9 | 0.0072 |
| SMC2 | Structural maintenance of chromosomes protein 2 | 2 | 2,8 | 0.119 |
| NDUAD | NADH dehydrogenase [ubiquinone] 1 alpha subcomplex subunit 13 | 2 | 2,7 | 0.1734 |
| TFE3 | Transcription factor E3 | 4 | 2,5 | 0.0361 |
| CKAP4 | Cytoskeleton-associated protein 4 | 9 | 2,4 | 0.00608 |
| KDM1A | Lysine-specific histone demethylase 1A | 6 | 2,4 | 0.0248 |
| SPT16 | FACT complex subunit SPT16 | 3 | 2,4 | 0.121 |

Top 20 nuclear TBL1XR1 interaction partners. After TBL1XR1 enrichment with an endogenous immunoprecipitation, interaction partners were identified using mass spectrometry. The ratio TBL1XR1/Flag shows the enrichment of the respective protein in the TBL1XR1 pulldown relative to the enrichment in the pulldown using Flag antibody. Statistical analysis was performed using a two-sided student's $t$ test. $n = 4$ wells per condition.

## Animal experiments

Mice were maintained under pathogen-free conditions on a 12 h light-dark cycle at 22 °C and 40–50% humidity with ad libitum access to water and regular chow diet (Altromin, Cat# 1314), unless stated otherwise. Animal handling and experimentation were performed in accordance with the institutional animal welfare officer. The necessary licenses were obtained from the state ethics committee and the government of Upper Bavaria (ROB-55.2-2532.Vet_02-16-136, ROB-55.2-2532.Vet_02-21-133, and ROB-55.2-2532.Vet_02-18-93).

All mice had C57BL/6 N background. Experiments were performed starting at 8–14 weeks of age. β-cell specific TBL/R1 knockout mice were generated by breeding TBL1X and TBL1XR1 floxed mice (Taconic Artemis, Cologne, Germany) and B6(Cg)-Ins1$^{tm1.1(cre)Thor}$/J mice (Jackson Laboratory)[25]. Inducible β-cell TBL/R1 knockout mice were generated by crossing TBL1X and TBL1XR1 floxed mice with mice carrying the tamoxifen-inducible Cre recombinase gene under the control of the rat insulin 2 promoter (RIP2Cre_ERT)[36]. RIP2Cre_ERT mice were kindly shared by the Ashcroft Lab. Knockout was induced by subcutaneous injection of 200 mg tamoxifen per kg body weight in corn oil. Tamoxifen-injected RIP2Cre_ERT negative littermates were used as controls. Homozygous db/db and heterozygous control mice were purchased from The Jackson Laboratory. TBL1X$^{fl/fl}$ TBL1XR1$^{fl/fl}$ mice were used for the aging cohort. This study used mostly male mice except for Supplementary Fig. S2a–d.

Body composition was determined using echo magnetic resonance imaging (EchoMRI, Echo Medical Systems, Houston). When stated HFD, mice were fed a high fat diet (60% calories from fat) starting at the age of 8 weeks (Research diets, Cat# D12492i).

For fasting and refeeding experiments, mice were fasted for 16 h and refed with a regular chow diet for 2 h. Blood was collected in EDTA-coated tubes by nicking the tail vein. Oral glucose tolerance tests (oGTT) were performed in 16 h-fasted mice by administration of 2 g glucose per kg body weight via oral gavage. If blood glucose levels exceeded the glucometer maximum, a blood glucose level of 600 mg/dL, which is the glucometer maximum, was assumed. Blood was collected into EDTA-

coated tubes by nicking the tail vein. Plasma was obtained by centrifugation at 2000 x $g$, 4 °C for 10 min. For intraperitoneal glucose tolerance tests (i.p. GTT), mice were fasted for 6 h and then injected with 2 g glucose per kg body weight intraperitoneally. For insulin tolerance tests (ITT), mice were fasted for 6 h and injected with 0.8 U insulin per kg body weight by intraperitoneal injection. Blood glucose levels were determined at the indicated time points.

Mice were sacrificed by cervical dislocation and subsequent decapitation for blood collection. Serum was obtained by centrifugation at 10000 x $g$, 4 °C for 10 min.

β-cell proliferation was determined by bromodeoxyuridine or 5-bromo-2'-deoxyuridine (BrdU) incorporation. Briefly, 100 mg per kg body weight BrdU was i.p. injected once a day for 3 consecutive days. Mice were subsequently sacrificed by cervical dislocation, and dissected pancreata were fixated in 4% histofix and later used for immunofluorescent imaging. Proliferative β-cells were defined as insulin$^+$ cells that co-stained with BrdU (Biotin anti-BrdU, Abcam, Cat# ab2284, RRID:AB_302944; Streptavidin (Cy3), Thermo Fisher Scientific, Cat# SA1010). Proliferative β-cells were determined as the ratio between proliferated β-cells and total β-cells.

## Immunofluorescent imaging

For histological analysis, pancreata were fixated in 4% histofix for 24 h and standardly processed for paraffin embedding (Tissue Tec VIP.6, Sakura Europe, Netherlands). Paraffinized pancreata were exhaustively cross-sectioned into 3-4 parallel, equidistant slices per case. Maintaining their orientation, the tissue slices were vertically embedded in paraffin. Immunohistochemical staining was performed on 3 μm consecutive slices using the following antibodies: anti-Insulin, Cell Signaling Technology, Cat# 3014; RRID:AB_2126503; anti-Glucagon, Takara Bio, Cat# M182; RRID:AB_2619627, goat anti-Rabbit IgG (H + L) Cross-Adsorbed Secondary Antibody, Alexa Fluor™ 750, Thermo Fisher Scientific, Cat# A21039, RRID:AB_2535710; goat anti-Guinea Pig IgG (H + L) Highly Cross-Adsorbed Secondary Antibody, Alexa Fluor™ 647, Thermo Fisher Scientific, Cat# A21450, RRID:AB_2535867). To

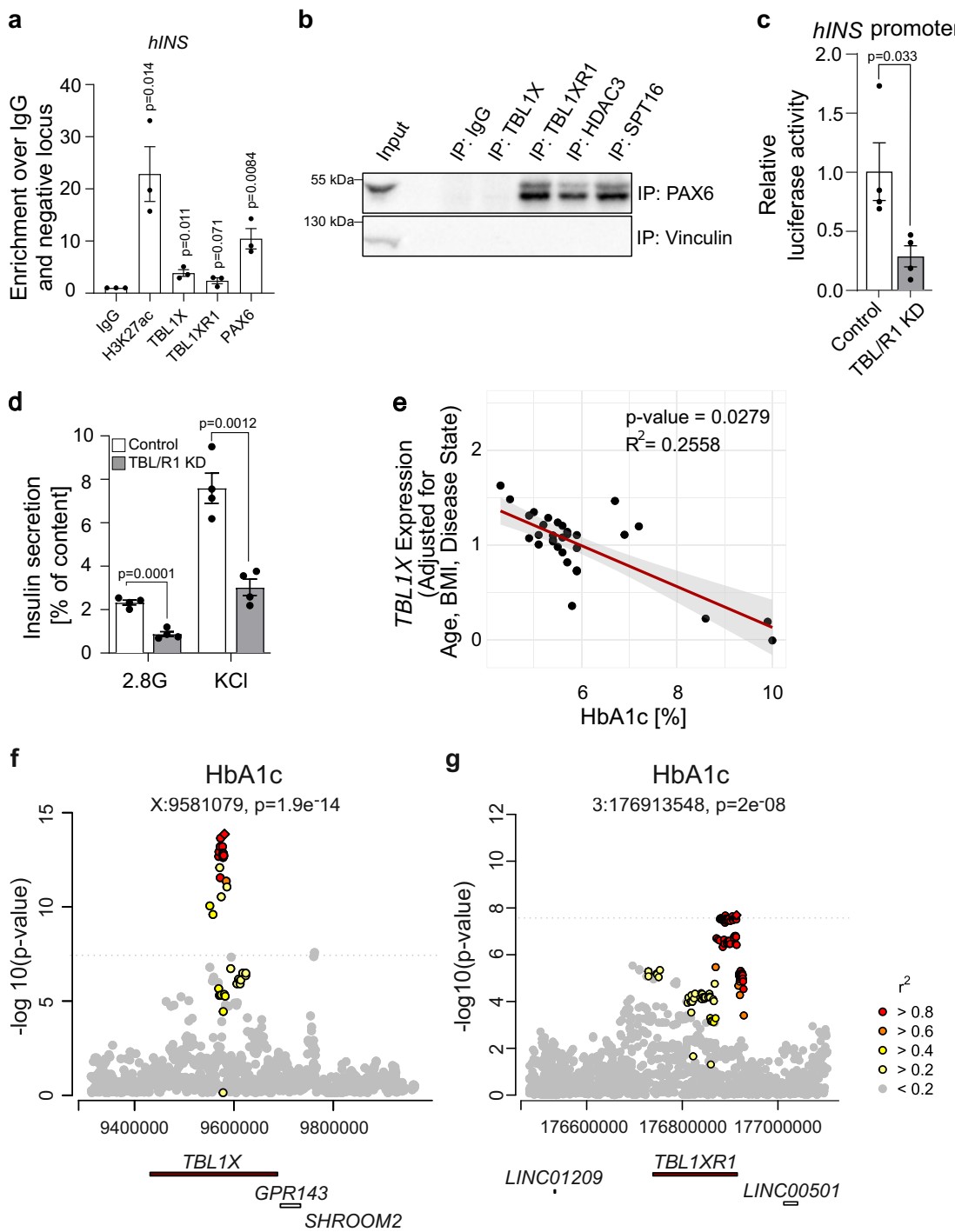

**Fig. 6 | TBL1X and TBL1XR1 requirement for β-cell function is conserved between rodents and humans. a** IgG (negative control), H3K27ac (positive control), TBL1X, TBL1XR1, and PAX6 ChIP-qPCR in EndoC-βH1 cells for the human *INS* locus. The enrichment is calculated over the negative locus and the IgG control. $n = 3$ individual experiments. **b** Endogenous TBL1X, TBL1XR1, HDAC3, and SPT16 were immunoprecipitated in EndoC-βH1 lysates with rabbit IgG as negative control. Whole cell lysates (Input) and immunoprecipitated proteins were immunoblotted using the indicated antibodies. Representative blot is shown for 4 independent experiments. **c** Human *INS* (*hINS*) promoter activity in INS1E cells upon TBL/R1 knockdown. Luciferase activity normalized to renilla luciferase. $n = 4$ wells/condition. **d** Insulin secretion relative to insulin content in EndoC-βH1 cells upon TBL/R1

knockdown. After glucose starvation, insulin secretion was stimulated using 2.8 mM glucose (2.8 G) and 2.8 mM glucose supplemented with KCl (KCl). $n = 4$ wells/condition. **e** Scatter plot depicting the relationship between HbA1c [%] and relative *TBL1X* mRNA levels determined by qPCR, adjusted for age, body mass index (BMI), and disease state. Each dot represents an individual data point. $n = 31$. **f, g** Manhattan plot showing an association between elevated HbA1c levels and single nucleotide polymorphisms in the genetic region of *TBL1X* (**f**) and *TBL1XR1* (**g**). Data are represented as mean ± SEM. The following statistical tests were applied: two-sided student's *t* test (**a**, **c**, **d**), and linear regression (**e**). Source data are provided as a Source Data file.

assess apoptosis, a terminal deoxynucleotidyl transferase dUTP nick end labeling (TUNEL) was performed using the In Situ Cell Death Detection Kit, TMR red (Roche, Cat# 12156792910). 1 slide containing two sets of stained tissue sections were scanned with an AxioScan.Z1 digital slide scanner (Zeiss, Jena, Germany) with a 20x magnification objective. Digital image analysis was performed using the Definiens® software. The α/β cell ratio was calculated based on the total α- and β-cell mass for each mouse. Total α- and β-cell mass were determined by:

$$Total\ \alpha - or\ \beta - cell\ mass = \frac{\alpha - or\ \beta - cell\ area}{total\ tissue\ area} \times pancreatic\ weight$$

### Total pancreas hormone content
Total pancreas hormone content was determined by acid ethanol extraction, followed by ELISA. Briefly, the dissected pancreata were placed into 5 mL ice-cold acid ethanol (0.2 M HCl in 70% ethanol), thoroughly homogenized, and left at −20 °C overnight. After the addition of another 5 mL of ice-cold acid ethanol and an incubation at − 20 °C overnight, debris was centrifuged at 4000 x g, 4 °C for 10 min and the supernatant was neutralized 1:1 using 1 M Tris-HCl pH 7.5 prior to determination of hormone content using an ELISA kit, as described below. Total pancreas hormone content was normalized to protein levels.

### Pancreatic islet isolation
Mice were sacrificed by cervical dislocation. For islet isolation, pancreata were inflated via the bile duct with 2 U/mL collagenase P in Hanks' Balanced Salt Solution (Sigma) supplemented with 1% BSA. Digestion was performed in a 37 °C water bath for 14 min. After a vigorous shaking, the reaction was stopped by adding ice-cold Hanks' Buffer containing 1% BSA. Islets were then washed and hand-picked 4 times. For adenovirus-mediated overexpression of TBL1XR1[17,61,62] murine pancreatic islets were cultured overnight in RPMI 1680 GlutaMAX (11 mM glucose) supplemented with 10% fetal bovine serum (FBS), 1% pen/strep. Islets were transduced at a multiplicity of infection (MOI) of 50 for 48 h.

Human donor (24 male, 7 female, average age 52,7 years (20-75 years)) organs were obtained with written consent and research ethics approval of the Human Research Ethics Board (HREB) at the University of Alberta (Pro00013094, Pro00001754). Perfusion was performed via the pancreatic ductal system with buffer containing Collagenase Gold 800 (VitaCyte, Indianapolis, IN) and Thermolysin (Roche Diagnostics, Mannheim, Germany), then digested with a Ricordi Islet Isolator (Biorep Diabetes, Miami, FL) and purified by density centrifugation[63]. Donor characteristics are described in Supplemental Data 2. Analyzed tissue samples are stored at the MacDonald Lab, University of Alberta. Please contact the corresponding author for further information.

### Enzyme-linked immunosorbent assay
Insulin and glucagon concentrations were determined using ELISAs according to the manufacturer's instructions. Plasma and total pancreas glucagon: Mercodia, Cat# 10-1281-01. Plasma insulin: Crystal Chem, Cat# 90080. Total pancreas insulin, glucose-stimulated insulin secretion and insulin content in EndoC-βH1 cells: ALPCO, Cat# 80-INSMS-E1.

### Cell culture
All cells were cultured at 37 °C, 5% CO$_2$, and 95% air atmosphere. INS1E cells (Lickert lab, RRID:CVCL_0351) were cultured in RPMI 1680 GlutaMAX medium (11 mM glucose) supplemented with 10% FBS, 1% pen/strep, 1 mM sodium pyruvate, 10 mM HEPES, and 50 μM β-mercaptoethanol. MIN6 cells (Seino lab, RRID: CVCL_4371) were cultured in DMEM high glucose (25 mM glucose) supplemented with 20% FBS and 1% pen/strep. EndoC-βH1 cells (Ämmälä lab, Human Cell

Design, RRID: CVCL_L909) were cultured in DMEM low glucose (5.5 mM glucose) supplemented with 2% BSA, 1% pen/strep, 2 mM glutamine, 10 mM nicotinamide, 50 μM β-mercaptoethanol, 5.5 μg/mL transferrin, 6.6 ng/mL sodium selenite. Knockdown of TBL1X and TBL1XR1 was induced using siRNA purchased from Dharmacon. For this, INS1E cells were transfected with 6 nM siTBL1X (Cat# J-096212-09-0005) and 25 nM siTBL1XR1 (Cat# J-096212-09-0005) using RNAiMAX (Invitrogen, Cat# 13778150, 1.5 μL/well) for 5 days. EndoC-βH1 cells were transfected with 25 nM siTBLX1 (Cat# L-012152-00-0005) and siTBL1XR1 (Cat# L-012927-00-0005) for 5 days, respectively to induce a TBL1X and TBL1XR1 knockdown. As control, a non-targeting control pool was used (Cat# D-001810-10-05). Overexpression of TBL1X and TBL1XR1[23] in INS1E cells was performed using expression constructs and Lipofectamine 2000 (Invitrogen, Cat# 11668027) for 48 h. Overexpression of TBL1X[23] in EndoC-βH1 cells was performed using expression constructs and Lipofectamine 2000 (Invitrogen, Cat# 11668027) for 48 h.

### Construction of reporter gene constructs and luciferase reporter assay
Murine *Ins2* and human *INS* promoter constructs were designed by the Ferreri Lab and purchased from Addgene[64]. The murine *Mafa* promoter construct was purchased from GeneCopoeia. Both, the murine *Ins2* and the murine *Mafa* reporter gene constructs were digested with HindIII/XbaI to separate the murine *Ins2* promoter from the pGL4.10 backbone containing the firefly luciferase and the murine *Mafa* promoter from the pEZX-PG02 backbone containing the gaussia luciferase. A subsequent ligation resulted in the insertion of the murine *Mafa* promoter into the pGL4.10 backbone. The construct was then transformed into DH5α competent cells (Invitrogen, Cat# 18265017) for plasmid production. Luciferase activity was assayed using the Dual-glo luciferase assay system (Promega, Cat# E2920). For this, INS1E cells were transfected with control siRNA or siRNA against TBL1X and TBL1XR1 for 72 h. After a medium change, cells were transfected with PAX6 expression constructs (purchased from Origene, BC011272), reporter gene constructs, and respective controls using Lipofectamine 2000 (Invitrogen, Cat# 11668027) for 48 h. For HDAC3 inhibition, expression vector transfected cells were exposed to either 10 μM BRD3308 (Sigma Aldrich, Cat# SML1639) or 5 μM RGFP966 (Cayman Chemicals, Cat# BYT-ORB636772) for 24 h. Firefly luciferase activity was normalized to renilla luciferase.

### Glucose stimulated insulin secretion (GSIS) assay
INS1E and EndoC-βH1 cells were glucose starved in Tanaka Robertson Krebs Ringer Buffer (118.5 mM NaCl, 25 mM NaHCO$_3$, 10 mM HEPES, 4.71 mM KCl, 2.54 mM CaCl$_2$, 1.19 mM KH$_2$PO$_4$, 1.19 mM MgSO$_4$) supplemented with 0.1% BSA for 1 h. Then, insulin secretion was induced by exposing the cells to the indicated glucose concentrations or glucose with 40 mM KCl for maximal insulin secretion for 1 h. Supernatant was then collected to determine insulin secretion. Insulin content was determined by harvesting the cells in ice-cold acid ethanol (0.2 M HCl in 70% ethanol) and subsequent sonication. For GSIS in pancreatic islets, freshly isolated islets were cultured overnight in RPMI 1680 GlutaMAX (11 mM glucose) supplemented with 10% fetal bovine serum (FBS), 1% pen/strep. Islets were then washed twice and subsequently glucose starved using Tanaka Robertson Krebs Ringer Buffer supplemented with 0.1% BSA and 1 mM glucose for 1 h. Insulin secretion was stimulated using indicated glucose concentrations and 40 mM KCl for 30 min, respectively. Supernatant was collected for the determination of insulin secretion. Islets were sonicated in ice-cold acid ethanol (0.2 M HCl in 70% ethanol) for insulin content determination.

### Quantitative PCR
RNA from TRIzol lysed human pancreatic islets was isolated via chloroform and a subsequent isopropanol precipitation. RNA from

pancreatic islets was isolated using the RNeasy Plus columns (Qiagen). RNA concentration was determined using a NanoDrop2000 spectrophotometer (Thermo Fisher). 1 µg of RNA was then reverse transcribed into cDNA using the High-Capacity cDNA kit (Applied Biosystems, Cat# 4368814) according to the manufacturer's recommendations. Real-time quantitative PCR was performed using the QuantStudio 6 Flex Real-Time PCR System (Applied Biosystems) and the Takyon™ Low Rox Probe Master Mix dTTP Blue (Eurogentec, Cat# UF-LPMT-B0701). RNA expression was analyzed using the ΔCt method. RNA from human and murine pancreatic islets were normalized to levels of TATA-box binding protein RNA. The following TaqMan probes were used: *Tbp* - Mm01277042_m1, *Tbl1x* - Mm01222202_m1, *Tbl1xr1* - Mm01283877_m1, *Ins1* - Mm01259683_g1, *Ins2* - Mm00731595_gH, *Nkx6.1* - Mm00454961_m1, *Slc2a2* - Mm00454961_m1, *Mafa* - Mm00845206_s1, *Pdx1* - Mm00435565_m1, *Pax6* - Mm00443081_m1, *Ucn3* - Mm00453206_s1, *ChgA* - Mm00514341_m1, *Ngn3* - Mm00437606_s1, *Ldha* - Mm01612132_g1, *Hk1* - Mm00439344_m1, *TBP* - Hs00427620_m1, *TBL1X* - Hs00959540_m1, *TBL1XR1* - Hs00226564_m1.

## Bulk RNA-sequencing

Bulk RNA-sequencing (RNA-seq) and data analysis was performed at Novogene (Cambridge, UK). After isolation, islets were immediately lysed in 500 µL TRIzol. RNA was extracted using the RNeasy Plus columns (Qiagen, Cat# 74134). After quality control procedures, the sequencing library was generated using NEBNext® Ultra™ RNA Library Prep Kit for Illumina® following the manufacturer's recommendations. Sequencing was performed using the NovaSeq6000 (Illumina) with 150 bp read-lengths in paired-end mode. Batch correction was not performed. RNA-seq reads were aligned to the mm39 mouse genome using the STAR software (version 2.6.1 d)[65]. No read trimming was performed. HTSeq (version 0.6.1) was used to count the read numbers mapped of each gene. Differential expression analysis between two groups (four biological replicates per condition) was performed using the DESeq2 R package (version 2 1.6.3). The resulting *p*-values were adjusted applying the Benjamini-Hochberg's method for controlling the false discovery rate (FDR). Differentially expressed genes were defined as genes with an adjusted *p*-value < 0.05 found by DESeq2. Shared functions among differentially expressed genes were identified using Kyoto Encyclopedia of Genes and Genomes (KEGG) for pathways. In addition, data was analyzed using Gene Set Enrichment Analysis (GSEA).

Cell type and gene expression deconvolution has been performed with BayesPrism (version 2.2.2)[66]. The reference for deconvolution is based on our own scRNA-seq published here, from which we only used the monohormonal cell types. The accompanying cell-type annotations are derived from our previous cell-type classification. Deconvolution of cell proportions and the gene expression matrix of β-cells have been computed following the package tutorial. From the gene expression matrix, we computed the activity of β-cell specific biological programs using GSVA (version 1.46.0)[67] on vst transformed pseudocounts[68]. Only endocrine cell types are shown in visualizations.

## Single-cell RNA-sequencing

For single-cell RNA-sequencing (scRNA-seq), islets were isolated from 4 mice per group and cultured overnight at 37 °C in RPMI 1680 GlutaMAX (11 mM glucose) supplemented with 10% FBS and 1% pen/strep. For each sample, 150 islets were pooled and dissociated using TrypLE Express (Gibco) for 20 minutes at 37 °C. Cells were then washed with PBS containing 2% FBS, filtered through a 40 µm Flowmi™ Cell Strainer (Bel-Art) and diluted to a final concentration of ~1000 cells/µl in PBS containing 2% FBS. The cell suspension was immediately used for scRNA-seq library preparation with a target recovery of 10000 cells.

Libraries were prepared using the Chromium Single Cell 3′ Reagent Kits v3.1 (10x Genomics, Cat# 1000268) according to the

manufacturer's instructions. Libraries were pooled and sequenced on an Illumina NovaSeq6000 with a target read depth of 50000 reads/cell.

Data preprocessing was done with the Cell Ranger pipeline (v 7.1.0), which aligned the reads, generated QC metrics, estimated the number of valid barcodes and created the count matrices. The command 'cellranger count' was executed with standard parameters, except that we adjusted the number of expected cells (10000) and the chemistry parameter (auto).

To remove the ambient profile from our data we identified empty droplets from unfiltered Cell Ranger output with "emptyDrops" from DropletUtils (version 1.18.1)[69] using distribution based lower thresholds (Ctrl: 400; KO: 175) and retaining cell containing droplets above the knee of the rank distribution.

For quality control and final data analyes, preprocessed count data from filtered Cell Ranger output were imported into Seurat (version 5.1.0)[70–72]. Filtering of cells during Seurat import was based on more than 2500 but less than 6000 expression features and mitochondrial fractions of reads < 10%. For normalization and variance stabilization, we used the modeling framework "sctransform"[73] following the Seurat v5 vignettes "vignettes/sctransform_vignette.Rmd". Ambiance removal was performed with "removeAmbience"[74] and based on previously identified ambient profiles. Integration of individual samples has been performed using anchor-based reciprocal PCA (RPCA) integration as described in Seurat's integration workflow "vignettes/seurat5_integration.Rmd". Dimensionality reduction plots are UMAP built from 30 PCs.

Identification of statistical clusters was based on nearest-neighbor graph construction from 30 PCs using the function "FindNeighbors" and clustering was performed with Seurat's internal function "FindClusters" with the parameter resolution = 2. It is based on a shared nearest neighbor (SNN) modularity optimization method by Waltman and van Eck[75].

Cell identity classification, including the identification of doublets or polyhormonal cells was performed in each sample independently. The canonical marker gene expression for the main endocrine and non-endocrine cell types[76] was extracted from the "RNA" slot and thresholded based on individual inspection of the data distributions ("Ppy" > 2.2, "Gcg" > 2.1, "Sst" > 2, "Ins1" > 2.0," Cd86 > 1, "Trac" > 0.1, "Cd34" > 0.1, "Vim" > 0.6, Krt19 > 0.5). Cells with high expression of exclusively one marker were used as "pure profile cells" in further analyses. To derive meaningful expression features for distinguishing cell types, the top 10 genes defining each cluster based on differential gene expression analysis have been identified for each of the 9 detected cell types within pure profile cells and concatenated together with the published marker genes[76]. Subsequently, centroids (average expression profiles) per cell type have been computed comprising scaled expression values of those discriminative marker genes from any of the pure profile cell types. Next, artificial mixed-type centroids were computed for each pairwise and triple-cell combination. In the final step, scaled transcriptomic marker profiles of all cells were individually correlated to all (pure and mixed) centroids and cells were classified according to the highest correlation. Cells with the highest correlations to mixed centroids were labeled as mixed or possibly polyhormonal based on which combination the artificially mixed centroid they were derived from. We additionally utilized two dedicated doublet detection algorithms: "DoubletFinder" (version 2.0.4)[77] and "Scrublet" (version 0.2.2)[78]. Thus, in total, we had inferences of mixed cell types from three different approaches and used a majority vote strategy to discriminate between mixed cell types and doublets. Mixed cells detected by more than one algorithm were classified as doublets and removed from further analyses. The remaining mixed endocrine cell types are regarded as suggestive polyhormonal cells and retain the initial cell type label from the maximum correlation to centroids. Quality control of classification was based on plausibility check in UMAPs and marker expression. For quantification and

visualization, we dismissed mixed cell types that comprise non-endocrine cells. For polyhormonal β-cell classification, accuracy will depend on the assumption that truly polyhormonal cells display an "in-between" expression profile of the respective monohormonal cell type markers. Limitations of our strategy are that polyhormonal cells with truly unique expression characteristics cannot be identified, and separation between polyhormonal and "doublet" cells might be imperfect. Differential gene expression analysis between knockout and control β-cells as well as marker identification for β-cell cluster 4 was done using DEseq2 statistical models[79] from the RNA assay with the function "FindMarkers" with parameter min.pct = 0.01. Since there are no biological replicates, we used individual cells for computing inference statistics. Pathway analysis with GO terms "molecular function" have been done using GSEA[80] within clusterProfiler[81]. Further pathway analyses were performed using Enrichr[82–84]. Visualizations were generated with "dotplot" and "emapplot" functions from the enrich plot package[85]. enrichplot: Visualization of Functional Enrichment Result. R package version 1.22.0, https://bioconductor.org/packages/enrichplot[86].

Computation of module scores was performed from the "SCT" Assay using Seurat's function "AddModuleScore" which utilizes the method by Tirosh et al.[87]. Inference statistics of module scores between β-cells from knockout and control animals were performed using non-parametric Wilcoxon tests with p-value adjustment for multiple testing by Holm. Differences in location parameters (median of the difference of samples from both experimental groups) are computed and interpreted as effect sizes. Biological programs were derived from Hrovatin et al.[32] and semantic similarity was computed between those terms with GOSemSim using the graph based Wang method[88] on the gene ontology "biological process" DAG and the best match average (BMA) method for aggregation.

Network generation and visualization has been performed with "igraph[89] ("The igraph software package for complex network research." InterJournal, Complex Systems, 1695. https://igraph.org.) and "ggraph" (ggraph: An Implementation of Grammar of Graphics for Graphs and Networks. R package version 2.2.1.9000, https://github.com/thomasp85/ggraph, https://ggraph.data-imaginist.com)[90]. Visualization of single-cell data and module scores was performed with build in Seurat functions "DimPlot" and "RidgePlot".

All statistical analyses involving scRNA-seq data have been perfomed in R (version 4.2.3) (R: A Language and Environment for Statistical Computing,R Core Team, R Foundation for Statistical Computing, Vienna, Austria, 2023, https://www.R-project.org).

### Chromatin immunoprecipitation (ChIP)

ChIP was performed in MIN6 and EndoβH1 cells as previously described[91]. Briefly, cells on a 15 cm dish (70% confluency) were crosslinked using 2 mM DSG (ProteoChem, Cat# C1104) and fixed in 1% formaldehyde (Thermo Fisher Scientific, Cat# 28906). Nuclei were isolated using a ChIP buffer (167.5 mM NaCl, 50 mM Tris HCl pH 7.5, 5 mM EDTA pH 7.5, 1% Triton X 100, 0.5% NP-40), supplemented with an EDTA-free protease inhibitor cocktail (Roche). Chromatin was sheared to a 0.2–1 kb size using a Diagenode Bioruptor then immunoprecipitated using 3 μg of the respective antibodies on a rotator at 4 °C overnight (anti-TBL1X, Proteintech, Cat# 13540-1-AP, RRID: AB_2199783; anti-TBL1XR1, Novus Biologicals, Cat# NB600-270, RRID: AB_10001343; anti-PAX6, Millipore, Cat# AB2237, RRID: AB_1587367; anti-H3K27ac, Abcam, Cat# ab4729, RRID: AB_2118291; anti-normal rabbit IgG, Cell Signaling Technology, Cat# 2729, RRID: AB_1031062). Sepharose™ Protein A/G beads (Rockland Immunochemicals, Cat# PAG50-00-0002) were added, and the bound chromatin was eluted (100 mM NaHCO3, 1% SDS) after thorough washing of the beads. DNA was then isolated using the MinElute PCR Purification Kit (Qiagen, Cat# 28004). DNA was then quantified using the PowerUp™ SYBR™ Green Master Mix (Thermo Fisher Scientific, Cat# A25742) in a QuantStudio 6

Flex Real-Time PCR System (Thermo Fisher Scientific) and normalized to a control locus and the IgG negative control. For this, the following primers were used: ChIP_*mFoxl2* fw – GCTGGCAGAATAGCATCCG, ChIP_*mFoxl2* rv – TGATGAAGCACTCGTTGAGGC; ChIP_*mIns2* fw – TGCAACTTCCTGGGGAATGAT, ChIP_*mIns2* rv – GCCCTGATGGCCT-GATGAA; ChIP_*hFOXL2* fw – GGATTGAACATACTTCGCGGC, ChIP_*h-FOXL2* rv – GGAGAACCAGACTGCAACCA; ChIP_*hINS* fw – TCCAGC TCTCCTGGTCTAATG, ChIP_*hINS* rv – TTGGTCGTCAGCCACCTCTTC.

### Protein extraction and immunoblotting

For protein extraction, the cells were lysed in ice-cold lysis buffer (150 mM NaCl, 50 mM Tris-HCl, pH 7.6, 1% Triton X-100, 0.5% Sodium deoxycholate, 0.1% SDS) supplemented with PhosSTOP (Roche) and an EDTA-free protease inhibitor cocktail (Roche) for 15 min on ice. 15 μg of protein extracts were separated using tris-glycine gels (Novex WedgeWell 8–16% Tris-Glycine Mini Gels; Thermo Fisher Scientific) and blotted on nitrocellulose membranes (Amersham), followed by membrane blocking and incubation with primary antibodies overnight at 4 °C (Anti-Vinculin, Abcam, Cat# 129002, RRID: AB_11144129; anti-TBL1X, Abcam, Cat# ab24548, RRID: AB_2199904; anti-TBL1XR1, Santa Cruz, Cat# sc-517365; anti-PAX6, Cell Signaling Technology, Cat# 60433, RRID: AB_2797599). After the secondary antibody incubation at room temperature for 1 h (goat anti-rabbit HRP, BioRad, Cat# 1705046, RRID: AB_11125757; goat anti-mouse HRP, BioRad, Cat# 1706516, RRID: AB_2921252; IgG Fraction Monoclonal Mouse Anti-Rabbit IgG, light chain specific, Jackson Immunoresearch Lab, Cat# 213-032-177, RRID: AB_2339251), protein bands were visualized using Amersham ECL Prime and the ChemiDoc MP Imaging System (BioRad).

### Interactome analysis

MIN6 cells were rinsed with PBS and lysed in ice-cold IP-lysis buffer (150 mM NaCl, 50 mM Tris/HCl, pH 6.8, 2 mM Na₃VO₄, 1 mM EDTA, 1 mM NaF, 1 mM DTT, 1% NP-40) supplemented with a PhosSTOP (Roche) and an EDTA-free protease inhibitor cocktail (Roche) for 15 min. After centrifugation, protein concentration in the lysates was determined using the BCA Protein Assay (Pierce). Then 6 mg of protein were incubated with 7 μg antibody (anti-TBL1X, Abcam, Cat# ab24548, RRID: AB_2199904; anti-TBL1XR1, Novus Biologicals, Cat# NB600-270, RRID: AB_10001343; anti-DYKDDDDK Tag, Cell Signaling Technology, Cat# 2368, RRID: AB_2217020) on a rotator at 4 °C overnight. Sepharose beads were added for 2 h on a rotator at 4 °C. Immunoprecipitated proteins were eluted in 1 x laemmli buffer by boiling at 95 °C for 10 min. 4 replicates were generated and analyzed for each pulldown. Proteins including agarose beads were subjected to tryptic digest applying a modified filter aided sample preparation (FASP) procedure as described[92,93]. Briefly, after protein reduction and alkylation using DTT and iodoacetamide, samples were denatured in UA buffer (8 M urea in 0.1 M Tris/HCl pH 8.5) and centrifuged on a 30 kDa cut-off filter device (PALL or Sartorius) and washed thrice with UA buffer and twice with 50 mM ammoniumbicarbonate (ABC). Proteins were proteolyzed for 2 h at room temperature using 0.5 μg Lys-C (Wako) and subsequently for 16 h at 37 °C using 1 μg trypsin (Promega). Peptides were collected by centrifugation and acidified with 0.5% trifluoroacetic acid (TFA).

LC-MS/MS analysis was performed on a Q-Exactive HF mass spectrometer (Thermo Scientific) online coupled to a nano-RSLC (Ultimate 3000 RSLC; Dionex). Peptides were accumulated on a nano trap column (300 μm inner diameter × 5 mm, packed with Acclaim PepMap100 C18, 5 μm, 100 Å; LC Packings) and then separated by reversed phase chromatography (nanoEase MZ HSS T3 Column, 100 Å, 1.8 μm, 75 μm X 250 mm; Waters) in a 80 min non-linear gradient from 3 to 40% acetonitrile (ACN) in 0.1% formic acid (FA) at a flow rate of 250 nl/min. Eluted peptides were analyzed by the Q-Exactive HF mass spectrometer equipped with a nano-flex ionization source. Full scan MS spectra (from m/z 300 to 1500) and MSMS fragment spectra were

acquired in the Orbitrap with a resolution of 60,000 or 15000, respectively, with maximum injection times of 50 ms each. Up to ten most intense ions were selected for HCD fragmentation depending on signal intensity (TOP10 method). Target peptides already selected for MS/MS were dynamically excluded for 30 s.

Spectra were analyzed using Progenesis QI software for proteomics (Version 3.0, Nonlinear Dynamics, Waters, Newcastle upon Tyne, U.K.) for label-free quantification as previously described[92]. All features were exported as Mascot generic file (mgf) and used for peptide identification with Mascot (version 2.4) in the SwissProt taxonomy mouse database (Release 2020_02, 17061 sequences). Search parameters used were: 10 ppm peptide mass tolerance and 20 mmu fragment mass tolerance, one missed cleavage allowed, carbamidomethylation was set as fixed modification, methionine oxidation and asparagine or glutamine deamidation were allowed as variable modifications. A Mascot-integrated decoy database search calculated an average false discovery of < 1%. GO annotation for biological processes was performed using the Cytoscape plugin ClueGO. Interaction partners with a Spectral Count > 2 and a unique peptide count > 1 were used for analysis. Analysis criteria comprised of a kappa score of 0.4 for pathway network connectivity and an inclusion of a minimum of 3 genes in the cluster with a GO tree interval range in between 3 and 8. Statistical analysis was performed with the Bonferroni step-down (pV correction) method, and terms with a significance < 0.05 were regarded as significant.

### Endogenous immunoprecipitation

MIN6, INS1E and EndoC-βH1 cells were lysed in an ice-cold IP-lysis buffer, as described above. After centrifugation, cell lysates were precleared using the SureBeads™ Protein G Magnetic Beads (BioRad, Cat# 161-4023). Precleared protein lysates were then incubated with 3 μg of the respective antibody (anti-TBL1X, Abcam, Cat# ab24548, RRID: AB_2199904; anti-TBL1XR1, Novus Biologicals, Cat# NB600-270, RRID: AB_10001343; anti-HDAC3, Abcam, Cat# ab32369, RRID: AB_732780; anti-SPT16, Proteintech, Cat# 20551-1-AP, RRID: AB_10700005; anti-rabbit (DA1E) mAb IgG XP® Isotype Control, Cell Signaling Technology, Cat# 3900, RRID: AB_1550038) on a rotating wheel at 4 °C overnight. The next day, SureBeads™ Protein G Magnetic Beads were added for 2 h. After washing, proteins were eluted in 2xSDS sample buffer by boiling at 95 °C for 10 min.

### Association assessment

We sought to test the associations of common variants located within *TBLX/ TBLXR1* genes and their ±500 kb flanking region with HbA1c and random blood glucose levels in the UK Biobank cohort study. The *TBLX* variants located on the X chromosome and their association with HbA1c and random glucose levels was tested in European participants from the UK Biobank study under application number 53639. The association analyses were performed using regenie[94] for both directly genotyped and confidently imputed variants, adjusting for age, sex, array types, and ten genetic principal components. We retrieved the genetic associations of *TBLXR1* variants from the publicly available summary statistics of two studies (HbA1c[95]; Random Glucose[96]) of the UK Biobank cohort. The genetic association analysis near *TBL1X* was done in European participants for HbA1c and random glucose (*n* = 364,509) in the UK Biobank, with 46% male and 8% T2D at baseline. Genetic associations near the *TBL1XR1* gene with HbA1c levels were obtained in 438,069 European participants of the UK Biobank[95], whereas random glucose level associations were obtained in a different study of 476,326 individuals without diabetes of European (*n* = 459,772) and other ancestries (*n* = 16,554)[96].

### Statistical analysis

Biological replicates were expressed as mean ± standard error of the mean (SEM). All statistical analyses were performed using GraphPad Prism version 10.2.3. Significance was tested using unpaired Student's *t* test, one-way analysis of variance (ANOVA), paired two-way ANOVA, and linear regression analysis as indicated in the figure legends. When comparing two variables, Tukey's or Šidák's *post hoc* tests were applied for correction. *p* < 0.05 was considered statistically significant.

### Limitation of the study

All data shown was generated using male mice. However, a subset of experiments was also performed in female mice, showing that also female TBL/RβKO mice develop hyperglycemia. Moreover, while α-cell mass remained unchanged, β-cell mass was significantly reduced determined by immunofluorescent staining of in paraffin embedded pancreas in female TBL/RβKO mice resulting in an elevated α/β-cell mass ratio. Lastly, abnormal features of islet cell distribution were also observed in female mice with α-cells scattered across the islet, as previously shown in male mice.

### Reporting summary

Further information on research design is available in the Nature Portfolio Reporting Summary linked to this article.

## Data availability

Bulk RNA-seq data are deposited to GEO with the dataset identifier GSE283723. scRNA-seq data are deposited to GEO with the dataset identifier GSE275357. The mass spectrometry data are deposited to the ProteomeXchange Consortium (http://proteomecentral.proteomexchange.org) via the PRIDE partner repository with the dataset identifier PXD054677. Source data for all figures are provided with this paper. Source data are provided in this paper.

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

## Acknowledgements

We thank Adriano Maida for his help and excellent management and organization skills. We thank Annette Feuchtinger and the core facility tissue pathology for their help in pancreas immunofluorescence staining. We thank Lisa Mehr for technical assistance. We thank the animal caretakers of Helmholtz Munich for their help in animal maintenance. We thank the whole team of the Helmholtz Research School for Diabetes for their continuous organizational support. Human islets for research were isolated at the Alberta Diabetes Institute IsletCore (http://www.bcell.org/adi-isletcore.html) with the assistance of the Human Organ Procurement and Exchange (HOPE) program, Trillium Gift of Life Network (TGLN), and other Canadian organ procurement organizations. We especially thank the organ donors and their families for their gift in support of diabetes research. P.E.M holds a Canada Research Chair in Islet Biology. C.J. is supported by the Deutsche Forschungsgemeinschaft Trans-Regio TRR333 BAT energy. A.W.H. and M.R. are funded through the Helmholtz Association - Initiative and Networking Fund. M.R. is funded by the European Research Council (ERC) under the European Union's Horizon 2020 research and innovation program (# 949017), German Diabetes Center (DZD) Next grant, EFSD/ Boehringer Ingelheim European research grant on "Multi-System Challenges in Diabetes", and an EFSD/ Novo Nordisk Foundation Future Leaders Award. M.R. and K.A.D. are funded by the German Research Foundation (DFG FOR5795-1_HyperMet).

## Author contributions

A.W.H. conceived and performed the majority of the experiments and co-wrote the manuscript. P.W. performed the scRNA-seq analysis, with instructions from A.W.H. and M.R. on polyhormonal cell visualization. C.J., K.M., J.G., R.T.E., and D.H. helped to perform experiments. M.S. performed the scRNA-seq library prep. W.G. did the UK biobank association study. C.W. and K.A.D. performed the regression analysis. I.S. and C.A. performed EndoC-βH1 experiments and helped with the interpretation, A.C.K. and S.M.H. performed the proteomics analysis. J.G.L. helped with the human islet experiment. H.L., F.A., M.B., P.E.M., and S.H. contributed valuable discussions and suggestions. M.R. conceived and supervised the study and wrote the manuscript. All authors edited the manuscript and agreed to the final version.

## Funding

## Competing interests

K.M., W.G., I.S., and C.A. are employees of Novo Nordisk. A.C.K. is employed at Bruker. The project was partly funded through a research grant through the Helmholtz Munich – Novo Nordisk strategic alliance to M.R. and S.H. The remaining authors declare no competing interests.
