## [Peer Review file · Nature Communications]

TBL1X/TBL1XR1 govern β -cell identity through a PAX6-containing gene regulatory network

Corresponding Author: Dr Maria Rohm

Version 0:

Reviewer comments:

Reviewer #1

(Remarks to the Author)

Summary of findings

Walth-Hummel et al. describe TBL1X and TBL1XR1 as regulators of beta cell identity and their role in diabetes progression. They perform a double knockout in mice and show a diabetic phenotype at 20 weeks as well as a change in beta/alpha mass at 5 weeks. They further characterize this knockout using RNA-seq, scRNA-seq, and a tamoxifen induced Cre. Finally, they use LC-MS/MS to identify binding partners that are characterized by promotor activity assays and ChIP. To identify relevance to humans, they repeated insulin promotor assays in human cell lines, compared with HbA1c expression, and identified GWAS signals near each gene. While this is a thorough characterization of the role of TBL1X and TBL1XR1 in diabetes development with a striking diabetic phenotype, there are many technical issues that will need to be addressed to make the findings and conclusions convincing.

Major comments

- Results from both bulk RNA-seq experiments are confounded by the alpha/beta ratio changes shown in figure 1 and figure 4. Because there is a large difference in the ratio of alpha and beta cells (specifically, fewer beta cells and more alpha cells), many of these results (with the possible exception of ECM pathways) could be explained just by the changes in cell type composition. For example, the disallowed genes discussed on line 155 are expressed by alpha cells or other islet cell types, so an increase in detection could purely reflect a higher proportion of alpha cells in the sample. Ideally, these experiments should be repeated with sorted beta cells. If experiments aren't repeated, any conclusions that reflect cell type composition should be removed or rephrased to indicate cell type composition changes rather than changes in beta cell identity and transcriptional regulation. It might be helpful here to use a bulk RNA-seq deconvolution method to attempt to better model these changes, however there are large caveats with this as many of the expected changes of dedifferentiation and transdifferentiation will be challenging to tease out from cell type composition changes. It is very challenging to identify polyhormonal cells by scRNA-seq because the ambient RNA for the islet hormones is incredibly high and it is challenging to computationally identify doublets. However, doublets were not computationally removed (line 694). Without this it is even more likely that these "polyhormonal" cells are artifacts. There are three additional pieces of information that are needed for a claim of polyhormonal cells.
 1. Is there a difference in the "% reads in cells" output by cellranger?
 2. How does this analysis change after running ``removeAmbience`` from ``DropletUtils`` or running ``scAR`` from ``scvi-tools``?
 3. Even with ambient RNA removal, it is still possible that a "polyhormonal" cell just had stochastically higher levels of background RNA. An alternative method of identifying these cells, such as immunostaining combined with z-stack confocal imaging and quantification, is required (although it doesn't appear that there are polyhormonal cells based on the images from example islets in this manuscript).
- Very little expression data is shown from the scRNA-seq which makes it impossible to gauge the quality of the data and interpret the data being shown. Please add in a heatmap, dot plot, or violin plots showing the expression of key genes associated with these cell types. At the very least, each of the described hormones should be visualized, but a broader characterization of these cells would be incredibly helpful.
- In addition to adding a heatmap/dot plot/violin plot for cell type markers, please also include this type of figure for the genes described in lines 189-191.
- Why were the islets cultured overnight before preparing the scRNA-seq libraries? This is certain to change the transcriptomic signature of the cells.
- All fastq files for bulk RNA-seq must also be deposited in GEO. The GSE for scRNA-seq and RNA-seq should be in their

own section titled “data availability”.

- More information (and code) needs to be provided for the identification of cell types. How were top 10 conserved markers genes (line 700) identified? What were the known marker genes (line 701)?

Minor comments

- The scale bar on the heatmaps is confusing. How is the log₂fold change per sample calculated? Normally, there is one log₂fold change value output at the end of the bulk DE pipelines (ie the mean of the KO compared to the mean of the control). It seems likely that these heatmaps are showing a z-score, which is the more traditional visualization of bulk RNA-seq expression.
- Line 179 “revealed a reduction in Beta-cells and enrichment in alpha, delta, and PP cells”. This should be clarified that it is only showing relative proportion and that actual increase in cell numbers cannot be determined (is the increase in delta cells just because there are fewer beta cells so the relative proportion increases but the actual cell number does not?).
- In figure 3c, how was this clustering done? Often with subcluster analysis, best practices include reidentifying variable features and rerunning PCA before generating a UMAP and clustering. Based on the figure, it looks like the PCA built on the full dataset was used.
- Line 719-720: UMAPs are generally best for visual representation of the data and should not be used to perform analysis. Instead of taking beta cells of interest based on UMAP coordinates, a cluster-based approach would be more appropriate.
- Add a reference to table S1 on line 191/192. Additionally, it was not clear from the citation (Weaning triggers a maturation step of pancreatic beta cells) how these pathways were generated.
- Figure 3f add beta cell maturity to either the title of the plot or the x axis.
- In figure 3f, it would be helpful to see all clusters separated by sample. Are there some cells in the KO that have higher levels of beta cell maturity? The ridge plot also hides the KO cells that have high expression of the module. The parameters should be adjusted so that the density plots don’t overlap.
- More description is required in the methods for bulk RNA-seq. What package was used for aligning? What version? How were genes counted? Was read trimming performed? What version of DESeq2 was used? Was any batch correction needed or performed? Why was no log₂fold change cutoff employed?
- Methods for the scRNA-seq analysis seem incomplete. How were the fastq files preprocessed? Was this using cellranger? What version and what parameters were used?
- Figure 1b, figure 4f-g: are these showing qPCR? This should be stated in the legend.
- Figure 4c: the # sign should be described in the legend.
- While most of the legends include the number of mice, some (like 4f) do not. Figure legends should be carefully checked.
- Figures 3 and S3, when describing the module score in the legend, “abundance” should be replaced with “module score”. This is a normalized score based on the expression of genes in different expression quantiles and does not directly represent abundance.
- Section “TBL1X and TBL1XR1 are required for PAX6 mediated insulin gene expression”: It isn’t clear from the text that this is in the INS1E cell line. It would be helpful to add this information in line 301.
- Figure 6e isn’t very convincing. The R² value is quite low and it is largely driven by the few points with high HbA1c values. Based on the data from the supplementary table, the HbA1c is significantly different between islets the non-diabetic and T2D donors. It would be appropriate to show a statistical test of TBL1X and TBL1XR1 expression in islets from ND and T2D donors. To further the analysis, it would be useful to build a linear model that includes the potential confounding factors such as BMI and age and include an interaction term for disease status.
- Line 427 states that the islets had a lower beta cell mass, but this was never directly characterized, only the ratio of alpha to beta cells.
- Line 493 seems to be missing a citation as I don’t think Tbl1x expression was shown in db/db animals in this study.
- Some of the references seem to be incorrect. For example, lines 41-46: lists genes that are mentioned as being regulated by PDX1, NKX6.1 and PAX6. The two references don’t talk about this regulation at all; line 183-184 the reference doesn’t appear to discuss diabetic islets, but instead focused on pancreatic development. All references should be thoroughly checked.
- The statistical tests listed in figure S1 don’t seem to match the panels, these should be corrected.

Reviewer #2

(Remarks to the Author)

The manuscript presented by Walth-Hummel et al identified transcription factors TBL1X/TBL1XR1 are important for the maintenance of functional beta cell. β -cell specific TBL1X/TBL1XR1 double-knockout mice and inducible knockout mice exhibited progressive β -cells dysfunction. Bulk and single-cell RNA sequencing show that β -cell identity is changed upon knockout. PAX6 is further identified as the interaction partner to regulate insulin release in both mouse and human beta cells. In humans, the islet gene expression of TBL1X correlates with HbA1c levels, and TBL1X and TBL1XR1 gene variants are associated with high HbA1c across the population. The data is well presented, and it is important to show these two transcription co-factors are important for beta cell’s function. Here are my questions:

1. Statistical results from different experiments should be included in all analyses and figures where possible, for example figure 1h and 1i, figure 6a and others.
2. Insulin expression should be directly measured in both human and mouse cells upon knockdown. In Figure 5h and 6d, it was shown that reduced insulin secretion upon knockdown. However, the level of insulin secretion involves many complicated processes which involves transcription, translation and secretion. Will overexpress TBL1X and TBL1XR1 increase insulin expression?

3. Loading/input control for all the samples should be shown figure 5c, 6b,
4. Does TBL1X/TBL1XR1 specifically expressed in beta cell only? The positive correlation has been showed between TBL1X from islets and HbA1C. However, it is not clear that if these two transcription factors can also be expressed from other islet cells. Cell type expression pattern of TBL1X/TBL1XR1 should be showed in both mouse and human data.
5. Were all the mice male or female? Are there any differences between them in this context?

Reviewer #3

(Remarks to the Author)

Summary of the key results

In this study, Walth-Hummel et al reported an intriguing discovery of Transducin β -like 1 x linked (TBL1X) and TBL1-related (TBL1XR1) gene function in β -cells. The team identified that β -cell specific TBL1XR1 knockout (KO) mice develop early onset hyperglycemia with reduced β cell mass and function, in contrast increased α cell ratio. Genomic analysis determined that TBL1XR1KO islets decreased the gene expression necessary for β -cell identity and increased gene expression features immature or progenitors. These characters are also observed postnatal deletion of TBL1XR1 gene in β -cells with tamoxifen-inducible β -cell specific TBL1XR1KO mice. The authors identified that TBL1XR1 interact with Pax6 and co-regulate insulin gene expression through its promoter activation. Finally, the authors showed that TBL1XR1 expression inversely correlate with the incidents of hyperglycemia in UK human patients. This study provides the detailed novel role and mechanistic insights of TBL1XR1 in murine and human β -cells. However, some questions remain to be elucidated.

Major points.

1. The authors demonstrated an increase in the α cell ratio in the islets of TBL/R β KO mice. However, it remains unclear whether this observed increase in the α cell ratio is due to a reduction in β cell mass or enhanced transdifferentiation from β to α cells in TBL/R β KO mice. To clarify this, the authors should provide the absolute mass of both β and α cells. Additionally, how did the authors calculate the α/β cell mass ratio for each individual mouse ($n = 4$)? Given the expected variability in islet size, structure, and α/β cell mass ratio among individual mice, how was this data normalized? It is recommended to present the distribution of the α/β cell ratio across individual mice rather than only showing the average for each mouse ($n = 4$).

2. Ins1 is not expressed in all β cells. Since the authors used Ins1-Cre to generate β -cell-specific TBL/RKO, they should confirm TBL/R deficiency in cluster 3 of TBL/R β KO islets to determine whether the mixed cell population observed is indeed due to TBL/R deficiency, as shown in the scRNA-seq data sets (Figure 3).

3. In Figure 2c, the authors observed that TBL/R β KO islets showed increased expression of progenitor marker genes, such as Ngn3, Sox9, and Sox17, which correspond to the state of endocrine progenitors, pancreatic progenitors, and endodermal cells, respectively. However, these markers do not appear clustered in Figure 3a and 3c in the scRNA-seq analysis of similar islet samples. Since these markers are also absent in cluster 3 (the mixed population), it raises the question: where have these progenitors gone?

It is further recommended to conduct cell type-specific differential gene expression analysis to clarify the impact of TBL/R deletion in Ins1-positive cells. Additionally, since the mixed population clusters not only near Ins1-positive cells but also near Gcg-positive cells, these two mixed populations should be analyzed separately.

4. In Figure 1, TBL/R β KO showed hyperglycemia without the use of an HFD. Why, then, did the authors use an HFD model for iTBL/R β KO in Figure 4? Did the authors observe hyperglycemia induction in iTBL/R β KO without HFD as well? Additionally, is there an explanation for why the hyperglycemic phenotype in iTBL/R β KO is milder and takes more than 3 months to develop, compared to the TBL/R β KO?

5. It has been shown that inducible Pax6 β KO rapidly develops hyperglycemia within approximately 3 weeks following tamoxifen injection, with plasma insulin levels reduced to less than 3 ng/ml (PMID: 27941241). Although iTBL/R β KO exhibits a significant reduction in pancreatic insulin content by 4 weeks after tamoxifen injection, normoglycemia persists for over 15 weeks. The authors should monitor plasma insulin levels weekly to determine if a reduction of circulated plasma insulin to below 3 ng/ml begins to appear only after 15 weeks.

6. The authors concluded that TBL1X and TBL1XR1 form a gene regulatory network with PAX6, while the reduction of Pax6 expression in iTBL/R β KO islets is approximately 10% (Fig. 4g). Is this reduction significant enough to influence PAX6-mediated regulation of other β -cell-specific genes, including Ins? If so, the authors should demonstrate how a ~10% reduction in Pax6 affects the expression of other β -cell-specific genes and whether PAX6 overexpression can rescue GSIS and gene expression in iTBL/R β KO islets.

7. It is missing that the detailed information about the cohort used for the genetic analyses in Figure 6f and 6g. Specifically, could author include the number of participants, their gender, genetic background, and whether they are patients with T1D, T2D, or glucose intolerance? Since the authors previously reported that adipose specific TBL/R β KO shows obesity and glucose intolerance phenotype (PMID: 23499424), the presented genetic analyses does not directly prove the role of TBL/R in human β -cells. In addition with random blood glucose level, the authors should explore whether the variants near the TBL1X gene are associated with hypoinsulinemia.

Minor points.

1. Please check citation. Some citation does not match what's the author stated in the manuscript. e.g. Line 456-457 Pax6

depletion ~ 44 (44. Abe, Y. et al. RANK ligand ~). Isn't this citation 34.?

2. Figure 6a. Please provide statistic analyses.

Version 1:

Reviewer comments:

Reviewer #1

(Remarks to the Author)

Walth-Hummel et al. have very thoroughly addressed my concerns. I only have one minor comment that Figure S3C is not very legible and could be expanded so that the text is not overlapping.

Reviewer #2

(Remarks to the Author)

I thank the authors for their substantial efforts in revision. The added experiments and clarifications have improved the manuscript overall, including the new mouse knockdown analyses and the overexpression studies in both mouse islets and a human β -cell model. However, the central mechanistic question remains insufficiently resolved: do TBL1X/TBL1XR1 regulate β -cell function by directly controlling insulin gene expression (INS/Ins1/Ins2), rather than indirectly via secretion machinery, cell state changes, or mixed endocrine-cell effects?

1. Human knockdown: Insulin gene expression is still not directly quantified in the human β -cell model upon TBL1X/TBL1XR1 knockdown. Given the stated hypothesis, the authors should measure INS mRNA in EndoC- β H1 (or comparable human β -cell model, human islet) under knockdown, alongside secretion readouts.
2. Overexpression interpretation: While overexpression experiments are appreciated, the results differ between mouse islets and EndoC cells, and INS expression in EndoC overexpression conditions is not directly measured. Please quantify INS mRNA in EndoC overexpression experiments and reconcile any species/model differences (including the possibility of non- β endocrine contributions in intact mouse islets).
3. Human islet cell-type specificity: The scRNA-seq analysis is helpful, and mouse data suggest endocrine-wide expression, increasing the need for β -cell-autonomous evidence. For human islets, the presented data are not sufficiently clear to assess specificity. Please provide clearer human evidence using higher-quality datasets and more quantitative visualization (e.g., dot plots with % expressing and mean expression).

Reviewer #3

(Remarks to the Author)

The authors have adequately addressed my major concerns as outlined previous round or revision.

Overall, the revised manuscript presents a comprehensive and mechanistically supported study demonstrating an important role of TBL1X/TBL1XR1 in maintaining β -cell identity and function.

Minor suggestion

While the mechanistic interaction with PAX6 is convincingly demonstrated, the term "gene regulatory network" may slightly overstate the scope of the global regulatory evidence presented. The authors may consider modestly tempering this wording in discussion to better align with the scale of mechanistic data provided.

Dear reviewers,

Thank you for the careful assessment of our manuscript and the constructive criticism, which has helped us to substantially improve the manuscript. We have prepared a point-by-point response to your queries below, with our answers in blue. New figures which have been added to the manuscript have been marked in the rebuttal letter with Fig. X or Fig. SX, whereas data shown exclusively to the reviewers has been marked with Fig. RX. All text changes have been marked in the main manuscript in yellow.

During the revision of the manuscript, two scientists have contributed substantially, and they are now added as co-authors:

Congcong Wang

Kenneth A. Dyar

Thank you,

Kind regards,

Alina Walth-Hummel and Maria Rohm

Reviewer #1

Walth-Hummel et al. describe TBL1X and TBL1XR1 as regulators of beta cell identity and their role in diabetes progression. They perform a double knockout in mice and show a diabetic phenotype at 20 weeks as well as a change in beta/alpha mass at 5 weeks. They further characterize this knockout using RNA-seq, scRNA-seq, and a tamoxifen induced Cre. Finally, they use LC-MS/MS to identify binding partners that are characterized by promotor activity assays and CHIP. To identify relevance to humans, they repeated insulin promotor assays in human cell lines, compared with HbA1c expression, and identified GWAS signals near each gene. While this is a thorough characterization of the role of TBL1X and TBL1XR1 in diabetes development with a striking diabetic phenotype, there are many technical issues that will need to be addressed to make the findings and conclusions convincing.

We thank the reviewer for the accurate summary and thorough assessment of our manuscript.

Major comments

- Results from both bulk RNA-seq experiments are confounded by the alpha/beta ratio changes shown in figure 1 and figure 4. Because there is a large difference in the ratio of alpha and beta cells (specifically, fewer beta cells and more alpha cells), many of these results (with the possible exception of ECM pathways) could be explained just by the changes in cell type composition. For example, the disallowed genes discussed on line 155 are expressed by alpha cells or other islet cell types, so an increase in detection could purely reflect a higher proportion of alpha cells in the sample. Ideally, these experiments should be repeated with sorted beta cells. If experiments aren't repeated, any conclusions that reflect cell type composition should be removed or rephrased to indicate cell type composition changes rather than changes in beta cell identity and transcriptional regulation. It might be helpful here to use a bulk RNA-seq deconvolution method to attempt to better model these changes, however there are large caveats with this as many of the expected changes of dedifferentiation and transdifferentiation will be challenging to tease out from cell type composition changes.

Indeed, bulk RNA-sequencing of any tissue including endocrine pancreas suffers from the intrinsic problem that multiple cell types within the tissue are pooled. Therefore, we completely agree with your statement “many of these [bulk seq] results could be explained just by the changes in cell type composition”. It is the reason why we have additionally performed the single-cell RNA-sequencing (scRNA-seq) to get some information regarding cell type composition and changes within each individual cell population. scRNA-seq is adequately suited to assess β -cell intrinsic changes to transcription (see comments regarding Fig. 3). We have re-phrased our conclusions on bulk RNA-seq to be more cautious:

Line 190: “When focusing on genes with specific importance for pancreatic islets (Lemaire et al. 2017; Pullen et al. 2010; Salinno et al. 2019; Salinno et al. 2021), we indeed observed a downregulation of the insulin and β -cell identity genes (e.g. *Mafa*, *Nkx6.1*, *Slc2a2*), while “disallowed” genes, progenitor genes, and genes associated with other cell types of the islets (e.g. *Ldha*, *Ngn3*, *ChgA*, *Gcg*, *Ppy*) were upregulated (Fig. 2c). **This can either mean gene expression is changed within the β -cells or cell type composition of the islets is changed (i.e. less β -cells and more non- β -cells).**”

We have further performed bulk RNA-seq deconvolution to try to better model changes to cell composition vs. within β -cells. When mapping the bulk RNA-seq data to single hormone expressing cells from our scRNA-seq results, we indeed find lower levels of Ins1^+ cells and slightly elevated levels of other islet hormone expressing cell types (Fig. S4c), in line with scRNA-seq data and immunohistochemistry. We find “maturity” signatures significantly downregulated within the Ins1^+ population, while “disallowed” and “dedifferentiated” signatures tend to be increased ($p=.07$ and $p=.13$, respectively) (Fig. S4d). While we are very cautious not to over-interpret these results due to the inherent difficulties of the methodology (limitation also mentioned in discussion, line 588), the data clearly demonstrate that the results of the bulk RNA-seq are linked to both, changes to islet cell composition as well as within β -cells.

- It is very challenging to identify polyhormonal cells by scRNA-seq because the ambient RNA for the islet hormones is incredibly high and it is challenging to computationally identify doublets. However, doublets were not computationally removed (line 694). Without this it is even more likely that these “polyhormonal” cells are artifacts.

We have now applied three doublet removal protocols (specified in the materials and methods section) with the aim of distinguishing between doublets and polyhormonal cells. We have removed all cells from the analysis that were detected as doublets by at least two of these methods. Please see UMAP below for reference (Fig. R1), as well as the provided R-markdown file (https://osf.io/9akx6/overview?view_only=f0f538c210fc473fbf1a4df428edf0ee). After doublet removal, cells with mixed hormone expression remained for both WT and KO animals, with a particular enrichment of mixed hormone cells around the β -cell population in KO islets. We consider these polyhormonal cells after doublet removal. All further analyses were done with the dataset after doublet removal.

We now show proportional numbers of monohormonal (Figure 3b, c) and polyhormonal cells (Figure S3d) after doublet removal. Of note, all further analysis (including all replies to the reviewers) was performed on cells identified as singlet.

Fig. R1: Cells identified as Singlet or Doublet. Cells were identified as Doublet when detected as doublet by at least two different methods. Doublets were removed from all further analyses.

- There are three additional pieces of information that are needed for a claim of polyhormonal cells.

1. Is there a difference in the “% reads in cells” output by cellranger?

Cell Ranger results indicate good quality libraries with low amount of ambient RNA:

522 Fraction Reads in Cells 88.3% WT

523 Fraction Reads in Cells 92.4% KO

To our understanding, the parameter “% reads in cells” is computed for the whole sample/library, not for individual cells. Therefore, we analyzed the related measure “number of transcripts per cell” (Fig. R2). Doublets would be expected to have higher numbers of transcripts per cell than the respective hormone expressing monohormonal cells, e.g. doublet $Ins1^+$ cells would have roughly double the transcript content compared to monohormonal $Ins1^+$ cells. This is not the case, so this analysis shows there are no major differences in transcript content between mono- or polyhormonal cells.

Even though it is unlikely based on the number of transcripts per cell, we cannot formally exclude that the cells termed polyhormonal after doublet removal still contain a small fraction of doublet cells – a limitation that is now acknowledged in the main text.

Fig. R2: Transcripts per cell in WT vs. KO sample. This analysis indicates that RNA content per cell is not systematically higher in cells identified as polyhormonal cells, compared to monohormonal cells, in both samples.

2. How does this analysis change after running `removeAmbience` from `DropletUtils` or running `scAR` from `scvi-tools`?

Thank you for the great suggestion. We have now run “removeAmbience” and performed all subsequent analyses of the scRNA-seq based on this updated dataset. “removeAmbience” did not change the key results of our work since we observed no major changes upon “removeAmbience”. The main effect was an optimization of the Ins1⁺ cell cluster. Please see the comparison after vs. before “removeAmbience” below (Fig. R3) for reference. All further analyses were done with the dataset after “removeAmbience”.

Fig. R3: Comparison of cluster identities before and after “removeAmbience”

3. Even with ambient RNA removal, it is still possible that a “polyhormonal” cell just had stochastically higher levels of background RNA. An alternative method of identifying these cells, such as immunostaining combined with z-stack confocal imaging and quantification, is required (although it doesn’t appear that there are polyhormonal cells based on the images from example islets in this manuscript).

This is a great suggestion. We have now analyzed paraffin embedded pancreas stained for insulin and glucagon from control (n=4) and TBL/R β KO (n=4) mice at the age of 20 weeks. 3-14 islets (> 50 μ M) were analyzed per mouse. Cells co-stained for insulin and glucagon were defined as polyhormonal. This analysis showed that polyhormonal cells are a feature of TBL/R β KO mice although we haven’t found polyhormonal cells in all islets and overall numbers were low, as expected. This approach aimed at demonstrating the presence of polyhormonal cells in knockout islets which strengthens our results obtained from the scRNA-seq. This data is shown in Fig. R4 and was added as Supplemental Fig. S3e. Of note, we here only quantified and analyzed Ins1⁺/Gcg⁺ cells. However, the number of polyhormonal events might be underestimated as we have not investigated other types of polyhormonal cells e.g. Ins1⁺/Sst or Ins1⁺/Ppy⁺, especially since the Ins1⁺/Ppy⁺ population is very prominent in our scRNA-seq analysis (Fig S3d).

Fig. R4: Identification of polyhormonal cells in pancreatic islets using immunofluorescence. (A) Representative image of a polyhormonal cell in a pancreas detected by immunofluorescence. (B) Quantification of polyhormonal cells in islets of control and TBL/R6KO mice. Statistical analysis was performed using the Mann-Whitney test. * $p < 0.01$.

- Very little expression data is shown from the scRNA-seq which makes it impossible to gauge the quality of the data and interpret the data being shown. Please add in a heatmap, dot plot, or violin plots showing the expression of key genes associated with these cell types. At the very least, each of the described hormones should be visualized, but a broader characterization of these cells would be incredibly helpful.

This is an excellent suggestion. We have now included additional data on our scRNA-seq analysis showing the expression of key islet genes (Fig. S3a, b); of the top 10 genes identifying pure profile glucagon, somatostatin, pancreatic polypeptide γ , and insulin-expressing cells, respectively (Fig. S3c); the total and proportional numbers of the main islet cell types (Fig. 3b,c, Fig. S3d); and the expression of genes associated with the functionality and maturity of β -cells (Fig. S3g, h).

- In addition to adding a heatmap/dot plot/violin plot for cell type markers, please also include this type of figure for the genes described in lines 189-191.

Thank you for the suggestion. We have now included a dotplot highlighting the top 10 up- and downregulated genes in the $Ins1^+$ cell cluster across β -cell clusters. Please see Fig. 3e.

- Why were the islets cultured overnight before preparing the scRNA-seq libraries? This is certain to change the transcriptomic signature of the cells.

Overnight processing of islets for library preparation improved the cell viability which is essential for a successful scRNA-seq. Our own experience (Hrovatin et al. 2023) and previous studies have shown that a cell viability $< 90\%$ would impair scRNA-seq results mainly due to contamination with RNA from dead cells (Yianni und Sharpe 2022). We tried different approaches to optimize viability (sequential sample processing, changing the buffer for islet isolation, overnight cultivation), with cultivating overnight showing the highest improvement on cell viability. Please see in the table below how the different approaches affected the viability.

Processing	Picking medium	Cultivation O/N	Viability [%]
After injection of all samples	HBSS + 1% BSA	No	83.7
After injection of all samples	RPMI + 10% FBS + 1% P/S	No	83.3

Sequential processing	HBSS + 1% BSA	No	73.1
Sequential processing	RPMI + 10% FBS + 1% P/S	No	79.6
After injection of all samples	HBSS + 1% BSA	Yes	77.4
After injection of all samples	RPMI + 10% FBS + 1% P/S	Yes	83.1
Sequential processing	HBSS + 1% BSA	Yes	95.4
Sequential processing	RPMI + 10% FBS + 1% P/S	Yes	96.6

Based on these preliminary experiments we have decided to culture the cells overnight to maximize viability. Since islets were isolated from normoglycemic mice, culture in RPMI containing 11 mM glucose is expected to change gene expression signatures only slightly, and in a similar manner between genotypes.

- All fastq files for bulk RNA-seq must also be deposited in GEO. The GSE for scRNA-seq and RNA-seq should be in their own section titled “data availability”.

We have now added an additional “data availability” section to the methods. In addition to the scRNA-seq and bulk RNA-seq, information on data deposition for the interactome screen was also added to this section.

Line 1093: “Bulk RNA-seq data are deposited to GEO with the dataset identifier GSE283723. scRNA-seq data are deposited to GEO with the dataset identifier GSE275357. The mass spectrometry data are deposited to the ProteomeXchange Consortium (<http://proteomecentral.proteomexchange.org>) via the PRIDE partner repository with the dataset identifier PXD054677.”

- More information (and code) needs to be provided for the identification of cell types. How were top 10 conserved markers genes (line 700) identified? What were the known marker genes (line 701)?

Thank you for highlighting this misleading phrasing. The “Top 10 conserved genes” were identified in a data-driven manner after classification of pure profile cells. To identify the pure profile cells, known markers were used as previously described (Matta et al. 2025). The Top 10 differentially expressed genes between the pure profile cells from our analysis were then used as marker genes. We have now changed the phrasing into “Top 10 genes defining each cluster identified by analysis” and included a heatmap for clarification (Fig. S3c). Code for cell type classification as R-Markdown can be evaluated and replicated with the linked files: https://osf.io/9akx6/overview?view_only=f0f538c210fc473fbf1a4df428edf0ee

Minor comments:

- The scale bar on the heatmaps is confusing. How is the log2fold change per sample calculated? Normally, there is one log2fold change value output at the end of the bulk DE pipelines (ie the mean of the KO compared to the mean of the control). It seems likely that these heatmaps are showing a z-score, which is the more traditional visualization of bulk RNA-seq expression.

We have calculated the log2fold change to the median for each sample. However, for a better understanding we have now calculated the z-score and changed the figures accordingly. For this, please see Fig. 2c and Fig. S2h-j.

- Line 179 “revealed a reduction in Beta-cells and enrichment in alpha, delta, and PP cells”. This should be clarified that it is only showing relative proportion and that actual increase in cell numbers cannot be determined (is the increase in delta cells just because there are fewer beta cells so the relative proportion increases but the actual cell number does not?).

Thank you for pointing this out. We do indeed see a small increase of Gcg⁺, Sst⁺, and Ppy⁺ cells upon TBL/RβKO. We now show total numbers in Fig. 3b and relative numbers in Fig. 3c of Ins1⁺, Gcg⁺, Sst⁺, Ppy⁺ cells. Fig. S3d now also shows total numbers of the identified polyhormonal cells.

- In figure 3c, how was this clustering done? Often with subcluster analysis, best practices include reidentifying variable features and rerunning PCA before generating a UMAP and clustering. Based on the figure, it looks like the PCA built on the full dataset was used

We had explored the option of subclustering following the mentioned best practices. This did not reveal any new informative cluster structures and therefore, the initial clustering was kept for further analysis. We included minor changes in the manuscript for clarification and removed the term “subclustering” throughout to avoid possible misunderstandings.

- Line 719-720: UMAPs are generally best for visual representation of the data and should not be used to perform analysis. Instead of taking beta cells of interest based on UMAP coordinates, a cluster-based approach would be more appropriate.

Thank you for this insightful suggestion. Selection of β-cells is now solely based on the computed Seurat clusters.

- Add a reference to table S1 on line 191/192. Additionally, it was not clear from the citation (Weaning triggers a maturation step of pancreatic beta cells) how these pathways were generated.

We have added the reference to Table S1 and included the correct citation regarding pathway generation (Hrovatin et al. 2023).

- Figure 3f add beta cell maturity to either the title of the plot or the x axis.

Please refer to Fig. 3f (now Fig. 3g) for the changes.

- In figure 3f, it would be helpful to see all clusters separated by sample. Are there some cells in the KO that have higher levels of beta cell maturity? The ridge plot also hides the KO cells that have high expression of the module. The parameters should be adjusted so that the density plots don't overlap.

We thank the reviewer for this question and the excellent suggestion to alter the graph appearance in a way that all data can be seen. We have now shown the ridge plots from Fig. 3f, now Fig. 3g (as well as Fig. S4a, b) in transparent colors to address this. We display the module scores for each cluster from the control and TBL/RβKO groups separately. With this we get a better characterization of the identified clusters and highlight the differences which occur due to the absence of TBL/R1.

- More description is required in the methods for bulk RNA-seq. What package was used for aligning? What version? How were genes counted? Was read trimming performed? What version of DESeq2 was used? Was any batch correction needed or performed? Why was no log2fold change cutoff employed?

We have added the requested details for the bulk RNA-seq analysis to the methods section and highlighted them. In short:

- What package was used for aligning? What version? STAR software, version 2.6.1d
- How were genes counted? HTSeq, version 0.6.1
- Was read trimming performed? No
- What version of DESeq2 was used? Version 2_1.6.3

- Was any batch correction needed or performed? No, not needed.
- Why was no log2fold change cutoff employed? We were interested in a general overview of changes in gene expression rather than in the quantitative changes.

- Methods for the scRNA-seq analysis seem incomplete. How were the fastq files preprocessed? Was this using cellranger? What version and what parameters were used?

Thank you for your suggestions. We have now included information on the preprocessing into the methods part.

- Figure 1b, figure 4f-g: are these showing qPCR? This should be stated in the legend.

Yes, qPCR was applied to determine gene expression shown in these figures. We have added this information to the respective figure legends and supplemental figure legends.

- Figure 4c: the # sign should be described in the legend.

The #-sign in Fig. 4c was used to show significance comparing the insulin content in the pancreas of mice 4 vs. 11 weeks post tamoxifen administration. We have added this information to the figure legend of Fig. 4.

- While most of the legends include the number of mice, some (like 4f) do not. Figure legends should be carefully checked.

We have visualized results from mouse experiments using bar graphs showing individual data points, with each point representing one mouse. We have, however, thoroughly checked our legends and added the number of mice for all experiments. In Fig. 4f, we show results from a qPCR experiment where the number of mice varies for some genes due to the limitation of the sample volume. Please see below the information on the numbers of mice used for each gene:

Fig. 4f, g: Relative mRNA expression determined by qPCR in pancreatic islets of iTBL/R β KO and control mice of *Tbl1x* and *Tbl1xr1* (f) and known genes of β -cell identity and function (g, left) and “disallowed” genes (g, right). Control mice *Tbl1x*, *Tbl1xr1*, *Ins1*, *Ins2*, *Nkx6.1*, *Slc2a2*, *Mafa*, *Pdx1*, *Pax6*, *Ucn3*: n=9, Control mice *ChgA*, *Ngn3*, *Ldha*, *Hk1*: n=8, iTBL/R β KO mice *Tbl1x*, *Ngn3*, *Ldha*, *Hk1*: n=7, iTBL/R β KO mice *Tbl1xr1*, *Ins1*, *Ins2*, *Slc2a2*, *Mafa* n=6, iTBL/R β KO mice *Nkx6.1*, *Pdx1*, *Pax6*, *Ucn3*, *ChgA*: n=5.

- Figures 3 and S3, when describing the module score in the legend, “abundance” should be replaced with “module score”. This is a normalized score based on the expression of genes in different expression quantiles and does not directly represent abundance.

Fig. 3 and Fig. S3 (now S4) were adjusted accordingly.

- Section “TBL1X and TBL1XR1 are required for PAX6 mediated insulin gene expression”: It isn’t clear from the text that this is in the INS1E cell line. It would be helpful to add this information in line 301.

We have added the information on used cell lines throughout.

- Figure 6e isn’t very convincing. The R2 value is quite low and it is largely driven by the few points with high HbA1c values. Based on the data from the supplementary table, the HbA1c is significantly different between islets the non-diabetic and T2D donors. It would be appropriate to show a statistical test of TBL1X and TBL1XR1 expression in islets from ND and T2D donors. To further the

analysis, it would be useful to build a linear model that includes the potential confounding factors such as BMI and age and include an interaction term for disease status.

We have now compared islet *TBL1X* and *TBL1XR1* gene expression levels from human non-diabetic (ND) and type 2 diabetic (T2DM) donors (Fig. R6). Comparing *TBL1X* and *TBL1XR1* gene expression levels using a student's t-test, no significant differences between ND and T2DM donors were observed. An explanation for this is that we only got limited access to T2DM donors (n=6) in comparison to ND donors (n=25) which skewed our statistics. This was also the initial reason for us to analyze this data using a Pearson's correlation, in which the natural variation in HbA1c in the physiologic range is accounted for.

Fig. R6: Gene expression measured by qPCR of *TBL1X* and *TBL1XR1* in human islets isolated from non-diabetic (ND, n=25) and diabetic (T2DM, n=6) donors.

We also carefully evaluated the necessity of an interaction term by incorporating the interaction between HbA1c and disease status (T2DM) in our regression analysis. However, our results indicate that the interaction term is not a statistically significant predictor. Moreover, disease status (T2DM) has already been included as an independent covariate in the model, which effectively accounts for its potential confounding effect. Given that HbA1c is a well-established biomarker reflecting glycemic control, it is highly correlated with T2DM diagnosis. The inclusion of both variables in the model without an interaction term ensures that their independent contributions to *TBL1X* expression are appropriately adjusted for, without introducing multicollinearity or unnecessary complexity.

Given that the interaction term (HbA1c & T2DM) does not provide additional explanatory power and lacks statistical significance, removing it enhances the model's interpretability and prevents overfitting while maintaining robust adjustment for potential confounders, including BMI, age, and disease status. Thus, we believe that the current model—adjusting for age, BMI, and disease status without an interaction term—appropriately accounts for relevant confounding factors while preserving model parsimony and interpretability.

A linear regression analysis was performed with HbA1c as independent variable and *TBL1X* and *TBL1XR1* gene expression as the dependent variable, respectively. The coefficient of determination ($R^2 = 0.2558$) and p-value = 0.0279 displayed in the figure indicate a significant negative association between HbA1c and *TBL1X* expression after adjusting for potential confounders, while no association between HbA1c and *TBL1XR1* expression was observed. The old graphs have been replaced with new graphs accounting for confounders (Fig. 6e, S7c).

- Line 427 states that the islets had a lower beta cell mass, but this was never directly characterized, only the ratio of alpha to beta cells.

This is an excellent point, and we have now in addition to the α/β -cell mass ratio also added α - and β -cell mass separately for TBL/R β KO mice and respective controls at the age of 5 and 20 weeks. See also comment 1 by reviewer 3, and Fig. S1m, n.

- Line 493 seems to be missing a citation as I don't think *Tbl1x* expression was shown in db/db animals in this study.

We show in Fig. S7a *Tbl1x* gene expression in islets of db/db mice in comparison to heterozygous litter mates, and our discussion refers to this data. We have added the reference to the figure in the discussion to avoid misunderstanding.

- Some of the references seem to be incorrect. For example, lines 41-46: lists genes that are mentioned as being regulated by PDX1, NKX6.1 and PAX6. The two references don't talk about this regulation at all.

We thank the reviewer for noticing these problems with citations, which occurred due to Endnote problems while switching between computers. We have now thoroughly checked the references and adjusted them accordingly.

- line 183-184 the reference doesn't appear to discuss diabetic islets, but instead focused on pancreatic development. All references should be thoroughly checked.

This is a very important point and we have now checked all the references thoroughly.

- The statistical tests listed in figure S1 don't seem to match the panels, these should be corrected.

The figure legends were thoroughly checked throughout the manuscript, and the statistical tests were matched with the panels in Fig. S1.

Reviewer #2

The manuscript presented by Walth-Hummel et al identified transcription factors TBL1X/TBL1XR1 are important for the maintenance of functional beta cell. β -cell specific TBL1X/TBL1XR1 double-knockout mice and inducible knockout mice exhibited progressive β -cells dysfunction. Bulk and single-cell RNA sequencing show that β -cell identity is changed upon knockout. PAX6 is further identified as the interaction partner to regulate insulin release in both mouse and human beta cells. In humans, the islet gene expression of TBL1X correlates with HbA1c levels, and TBL1X and TBL1XR1 gene variants are associated with high HbA1c across the population. The data is well presented, and it is important to show these two transcription co-factors are important for beta cell's function.

We thank the reviewer for the positive evaluation of our study.

- Statistical results from different experiments should be included in all analyses and figures where possible, for example figure 1h and 1i, figure 6a and others.

Statistical results are displayed as significance stars (*), with no stars stating that no significance was reached. This is the case for Fig 1h and i. We now additionally added "ns" as indication when no significance was reached. Statistical results for Fig. 6a and Fig. S5b (now S6b) were performed and added to the respective figures and explained in the figure legends.

- Insulin expression should be directly measured in both human and mouse cells upon knockdown. In Figure 5h and 6d, it was shown that reduced insulin secretion upon knockdown. However, the level of insulin secretion involves many complicated processes which involves transcription, translation and secretion.

While the knockdown of TBL/R1 reduced *Ins2* gene expression in the rodent INS1E β -cell line, *Ins1* gene expression remained unchanged (Fig. R8). We have included the *Ins2* gene expression upon TBL/R1 knockdown in INS1E cells into our main figures (Fig. 5h). Thank you for raising this question.

As stated above, insulin secretion is controlled on different levels and involves an interplay of many processes. Moreover, *in vitro* models of pancreatic β -cells such as INS1E, MIN6, or EndoC- β H1 cells are established models for mechanistic studies, however it is also known that not all processes that take place *in vivo* are conserved. Moreover, physiological amplification of insulin secretion, which requires nutrient availability and β -cell nutrient metabolism was not considered in this experimental set up. Thus, we cannot exclude that manipulation of TBL/R1 expression can also have other effects on pancreatic β -cells apart from insulin gene regulation.

Fig. R8. *Ins1* and *Ins2* gene expression in INS1E cells measured by qPCR upon knockdown of TBL/R1.

- Will overexpress TBL1X and TBL1XR1 increase insulin expression?

TBL1XR1 overexpression (OE) was induced in primary mouse islets using an adenovirus. While *Ins1* and *Ins2* gene expression was unchanged (trend to increase for *Ins2*) upon TBL1XR1 OE, we observed an increase in basal and glucose stimulated insulin secretion upon TBL1XR1 OE (Fig. R9), which validates our previous observations, where the OE of TBL1XR1 increased insulin secretion in INS1E cells (please see Fig. 5i). TBL1X OE was induced in EndoC- β H1 cells using plasmid DNA. This caused a very mild increase in insulin secretion in the basal state (Fig. S6g, h)

Fig. R9: Adenovirus-mediated *TBL1XR1* overexpression in primary mouse islets: overexpression shown on protein levels with western blot; *Ins1* and *Ins2* gene expression measured by qPCR. Insulin secretion upon stimulation with 2.8 mM glucose (2.8G), 16.8 mM glucose (16.8G) or 2.8mM glucose + KCl (KCl).

We have now incorporated these results into the manuscript, Fig. 5 and S6.

- Loading/input control for all the samples should be shown figure 5c, 6b.

In these experiments we generated a large pool of protein lysate as input sample. From this pool, an equal amount of protein was distributed for the respective pulldown experiments. The shown input originated from this pool. Thus, we don't have the input for each pulldown but rather a general input (shown in the immunoblots) from which the pulldowns were performed. This way we minimize technical error.

- Does *TBL1X*/*TBL1XR1* specifically expressed in beta cell only? The positive correlation has been showed between *TBL1X* from islets and HbA1C. However, it is not clear that if these two transcription factors can also be expressed from other islet cells. Cell type expression pattern of *TBL1X*/*TBL1XR1* should be showed in both mouse and human data.

Thank you for pointing this out. *TBL/R1* are expressed in both β -cells and other islet cell types in both mice and human fetal pancreas (please see data below from our scRNA-seq data and (Olaniru et al. 2023), respectively, Fig. R10). For the current manuscript, we have focused on their role in β -cells only, but cannot rule out their physiological role in other islet cell types. We have added a statement regarding *TBL/R1* expression patterns in the discussion, lines 658-662.

Our data indeed show the correlation between HbA1c and *TBL1X* expression in islets. It is technically very difficult to separate the β -cells from the other islet cell types using samples from human donors, and sample material is extremely limited. Since β -cells represent a large proportion of islet cells, we here used the islet *TBL1X* expression as proxy and assumed that most of the expression signal would come from the β -cells, which indeed has the limitation that we cannot assess the relative contributions of other cell types. Due to limitations in sample material we are unfortunately unable to assess the individual cell type contribution to gene expression changes for this study. We have acknowledged this limitation in the manuscript.

Fig. R10: *TBL1X* and *TBL1XR1* gene expression in mouse islets assessed by scRNA-seq (our data) and in human fetal pancreas from PMID: 36513063.

- Were all the mice male or female? Are there any differences between them in this context

A basic, general characterization was also performed in female mice, however not as thoroughly, as the females displayed a similar but milder phenotype. Please see below that female *TBL/R β KO* mice showed significantly higher blood glucose levels at the age of 20 weeks in comparison to control littermates. Moreover, although α -cell mass was unchanged upon *TBL/R β KO* in females, β -cell mass was significantly reduced, and therefore the α/β -cell mass ratio was significantly increased. Lastly, pancreatic islets from female *TBL/R β KO* mice show comparable features to male *TBL/R β KO* mice with α -cells scattered across the islet. We have now included these results as Fig. S2a-d. We have also included a statement on the mouse sex in the methods section.

*Fig. S2a-d. a, Blood glucose levels of 20 week old female TBL/RβKO (n=3) and control (n=4) mice. b, Pancreatic α- (left) and β-cell mass (right) of female TBL/RβKO (n=3) and control (n=4) mice at 20 weeks of age. c, α/β-cell mass ratio of pancreatic islets from female TBL/RβKO (n=3) and control (n=4) mice at 20 weeks of age. d, Representative immunofluorescent staining of insulin⁺ (blue, β-cells) and glucagon⁺ (red, α-cells) cells of paraffin embedded pancreas from female TBL/RβKO and control mice at 20 weeks of age. Data are represented as means ± SEM. Each dot represents one mouse. Statistical analysis was performed using a student's t-test. *p < 0.05, **p < 0.01.*

Reviewer #3

In this study, Walth-Hummel et al reported an intriguing discovery of Transducin β-like 1 x linked (TBL1X) and TBL1-related (TBL1XR1) gene function in β-cells. The team identified that β-cell specific TBL1XR1 knockout (KO) mice develop early onset hyperglycemia with reduced β cell mass and function, in contrast increased α cell ratio. Genomic analysis determined that TBL1XR1KO islets decreased the gene expression necessary for β-cell identity and increased gene expression features immature or progenitors. These characters are also observed postnatal deletion of TBL1XR1 gene in β-cells with tamoxifen-inducible β-cell specific TBL1XR1KO mice. The authors identified that TBL1XR1 interact with Pax6 and co-regulate insulin gene expression through its promoter activation. Finally, the authors showed that TBL1XR1 expression inversely correlate with the incidents of hyperglycemia in UK human patients. This study provides the detailed novel role and mechanistic insights of TBL1XR1 in murine and human β-cells. However, some questions remain to be elucidated.

We thank the reviewer for the thorough assessment of our manuscript.

Major points:

- The authors demonstrated an increase in the α cell ratio in the islets of TBL/RβKO mice. However, it remains unclear whether this observed increase in the α cell ratio is due to a reduction in β cell mass or enhanced transdifferentiation from β to α cells in TBL/RβKO mice. To clarify this, the authors should provide the absolute mass of both β and α cells. Additionally, how did the authors calculate the α/β cell mass ratio for each individual mouse (n = 4)? Given the expected variability in islet size, structure, and α/β cell mass ratio among individual mice, how was this data normalized? It is recommended to present the distribution of the α/β cell ratio across individual mice rather than only showing the average for each mouse (n = 4).

Absolute α- and β-cell mass was determined but due to space limitations we initially decided to show the ratio only. We can indeed see an increase in α-cell mass over time, suggesting that β-cells might have transdifferentiated into α-cells or that β-cells lacking TBL/R1 co-express glucagon and are therefore identified as α-cells. We have now also included the total α- and β-cell mass into the manuscript. Please see Fig. S1m and n for this.

The α/β cell ratio was calculated based on the total α - and β -cell mass for each mouse. α - and β -cell mass were determined by:

$$\text{Total } \alpha\text{- or } \beta\text{-cell mass} = \frac{\alpha\text{- or } \beta\text{-cell area}}{\text{total tissue area}} \times \text{pancreatic weight}$$

Thus, the α - and β -cell mass was normalized to tissue area and pancreas weight. This information has been added to the methods section.

- Ins1 is not expressed in all β cells. Since the authors used Ins1-Cre to generate β -cell-specific TBL/RKO, they should confirm TBL/R deficiency in cluster 3 of TBL/R β KO islets to determine whether the mixed cell population observed is indeed due to TBL/R deficiency, as shown in the scRNA-seq data sets (Figure 3).

Thank you for this suggestion. We have indeed also tried to confirm TBL/R1 knockout in β -cells using our sequencing datasets. However, due to technical limitations of the scRNA-seq, we cannot detect the knockout, but rather observe an upregulation of TBL/R1 gene expression. The reason is that in our model, a loss of function variant is generated through the deletion of exon 5. The absence of a functional TBL/R1 protein results in an upregulation of TBL/R1 gene expression. We have previously observed this using different Taqman probes in a qPCR where a probe targeting exon 2 and 3 shows an upregulation while the probe targeting exon 5 shows a downregulation of *Tbl1x* expression in islets of mice bearing a β -cell specific TBL1X knockout (Fig. R11).

Fig. R11: *Tbl1x* gene expression in mouse islets of WT and KO mice measured by qPCR with primers targeting exon 2/3 (left) or exon 5 (right).

In our scRNA-seq approach, not the whole transcript but only the 3' is used for sequencing annotation. Since the 3' of our TBL/R1 loss of function protein is intact we here observe an increased expression of the TBL/R1 genes in the islets of our knockout mice, rather than a decrease (please see below), and this increased expression overlaps with β -cells cluster 4 (Fig. R12). The sequencing length does not allow us to identify exon 5 deletion. Therefore, we cannot use the TBL/R1 deficiency as identifier for β -cells in our dataset. Of note, diminished TBL1X/R1 gene expression (i.e. exon 5 depletion) was confirmed in pancreatic islets of knockout mice (Fig. 1b).

Fig. R12: UMAP visualizing *Tbl1x* and *Tbl1xr1* gene expression by scRNA-seq

- In Figure 2c, the authors observed that TBL/R β KO islets showed increased expression of progenitor marker genes, such as *Ngn3*, *Sox9*, and *Sox17*, which correspond to the state of endocrine progenitors, pancreatic progenitors, and endodermal cells, respectively. However, these markers do not appear clustered in Figure 3a and 3c in the scRNA-seq analysis of similar islet samples. Since these markers are also absent in cluster 3 (the mixed population), it raises the question: where have these progenitors gone?

Thank you for this question. We now checked for the progenitor markers mentioned in Fig. 2 (bulk seq) in the β -cell population from scRNA-seq, and found that *Ngn3*, *Cd81*, and *Sox9* were strongly upregulated upon TBL/R1 knockout, whereas *Sox2* and *Sox17* were not detected (please see table below). Gene expression changes either did not reach significance (*Ngn3*, *Sox9*) or were not found within the top 10 genes that were displayed in Fig 3e. This data highlights the limitation of scRNA-seq, namely low coverage of low-abundance genes.

	avg_log2FC	p_val_adj
Ngn3	4,9822947	ns
Cd81	1,64138993	1,47E-05
Sox9	3,90798633	ns

To better address this question as well as the following (regarding DEG of β -cells from scRNA-seq) we have now included data to better characterize the β -cell population and knockout-induced changes specifically in this population, please see our reply to the next comment and Fig. S3c.

- It is further recommended to conduct cell type-specific differential gene expression analysis to clarify the impact of TBL/R deletion in *Ins1*-positive cells. Additionally, since the mixed population clusters not only near *Ins1*-positive cells but also near *Gcg*-positive cells, these two mixed populations should be analyzed separately.

Thank you for this suggestion. We now further investigated changes in gene expression upon TBL/R1 knockout cell type specifically. To this end, we have performed a pathway analysis of the top ($\log_2FC > 0.5$ and $padj > 0.05$) up- and downregulated genes in the *Ins1*-positive cluster. Using REACTOME, we found a significant enrichment of “beta-cell gene expression” and “beta-cell development” genes in the DOWN-regulated subset of genes, and enrichment of “stress-

response” genes in the UP-regulated genes. Genes with the highest significance for these pathways are mentioned in the main text. Further, we now include a representation of a subset of changed genes in a dotplot, highlighting their expression across different β -cell clusters (Fig. 3e). These new analyses highlight that changes in β -cell gene expression mostly drive changes in total gene expression as shown in Fig. 2.

With regards to the mixed population clusters, our data indicate that total numbers of these cells are very low overall (see Fig. S3d). This means that a differential expression analysis would rely on too few cells to yield meaningful results, and hence we would prefer not to show such an analysis in order to not encourage over-interpretation of the results.

- In Figure 1, TBL/R β KO showed hyperglycemia without the use of an HFD. Why, then, did the authors use an HFD model for iTBL/R β KO in Figure 4? Did the authors observe hyperglycemia induction in iTBL/R β KO without HFD as well? Additionally, is there an explanation for why the hyperglycemic phenotype in iTBL/R β KO is milder and takes more than 3 months to develop, compared to the TBL/R β KO?

In (Swisa et al. 2017), two daily doses of 8 mg Tamoxifen resulted in the PAX6 knockout. In another study (Gutiérrez et al. 2017), a dose of 100 mg/kg body weight was given *via* i.p. injection every other day for 3 days to achieve a knockout. In our study, we only used a single injection with 200 mg/kg body weight. With this approach, we aimed for a more physiological representation of the phenotype where we don't completely delete TBL/R1 but rather only reduce their expression as it is observed in diabetic humans or db/db mice. Of note, a pancreatectomy of ~60% is not necessarily inducing hyperglycemia (Thisted et al. 2020). So remaining functional β -cells can compensate for the loss of the majority of β -cells under unchallenged conditions. Therefore, in this study it was necessary to challenge the mice using a HFD to develop hyperglycemia while other characteristics of a TBL/R β KO (reduced insulin content, altered islet morphology) were already present 4 weeks post tamoxifen injection.

- It has been shown that inducible Pax6 β KO rapidly develops hyperglycemia within approximately 3 weeks following tamoxifen injection, with plasma insulin levels reduced to less than 3 ng/ml (PMID: 27941241). Although iTBL/R β KO exhibits a significant reduction in pancreatic insulin content by 4 weeks after tamoxifen injection, normoglycemia persists for over 15 weeks. The authors should monitor plasma insulin levels weekly to determine if a reduction of circulated plasma insulin to below 3 ng/ml begins to appear only after 15 weeks.

We measured fasting blood glucose and insulin levels 4 weeks, 17 weeks, and 24 weeks after tamoxifen injection (Fig. R13). We observed that over time fasting blood glucose levels were increasing in iTBL/R β KO mice which is in accordance with the weekly determined random blood glucose levels (please see Fig. 4b). In line with the significantly reduced blood glucose levels 4 weeks after tamoxifen injection, we also see increased fasting insulin levels. Although 24 weeks after tamoxifen injection fasting blood glucose levels were significantly increased in iTBL/R β KO, this was not reflected in fasting insulin levels. This suggests that a glucose stimulus is required to reveal the impairments in the insulin response, as previously shown during the oGTT (Fig. 4d, e), and that the remaining functional β -cells can compensate for the dysfunctional knockout β -cells to a large extent, until challenged.

Fig. R13: Fasting blood glucose and insulin levels in iTBL/R β KO mice over time.

- The authors concluded that TBL1X and TBL1XR1 form a gene regulatory network with PAX6, while the reduction of Pax6 expression in iTBL/R β KO islets is approximately 10% (Fig. 4g). Is this reduction significant enough to influence PAX6-mediated regulation of other β -cell-specific genes, including *Ins*? If so, the authors should demonstrate how a ~10% reduction in Pax6 affects the expression of other β -cell-specific genes and whether PAX6 overexpression can rescue GSIS and gene expression in iTBL/R β KO islets.

This is a very important point as indeed PAX6 is an essential regulator of β -cell function. We have induced a ~15% reduction of PAX6 gene expression in EndoC β H1 cells using siRNA. Downregulation of PAX6 expression by 15% did not change the expression of any of the analyzed β -cell identity genes or disallowed genes (Fig. R14) thus suggesting that observed effects in the iTBL/R β KO mice stem from the absence of TBL/R1 (and subsequently a change in PAX6 function or binding to transcription co-factors, as shown in Fig. 5f) rather than from changes in *Pax6* expression.

Fig. R14: Relative mRNA levels of PAX6 and islet hormone genes or identity genes in EndoC β H1 treated with Control or PAX6 siRNA.

It indeed is a great suggestion to overexpress *Pax6* in pancreatic islets from iTBL/R β KO mice to see a possible rescue of the phenotype. We have tried to overexpress *Pax6* in pancreatic islets from iTBL/R β KO mice using plasmid DNA transfection and although we tried multiple different approaches, the overexpression was not successful due to technical issues. This is also to us very unfortunate as this experiment was a great suggestion. Data from INS1E cells indicate PAX6 OE indeed activates *Ins2* promoter activity, and this is completely blunted by TBL/R1 knockdown (Fig. 5e, g)

- It is missing that the detailed information about the cohort used for the genetic analyses in Figure 6f and 6g. Specifically, could author include the number of participants, their gender, genetic background, and whether they are patients with T1D, T2D, or glucose intolerance? Since the authors previously reported that adipose specific TBL/R β KO shows obesity and glucose intolerance phenotype (PMID: 23499424), the presented genetic analyses does not directly prove the role of TBL/R in human β -cells. In addition with random blood glucose level, the authors should explore whether the variants near the TBL1X gene are associated with hypoinsulinemia.

The genetic association analysis near TBL1X, located on chromosome X, was done in European participants for HbA1c and random glucose (n=364,509) in the UK Biobank, with 46% male and 8% T2D at baseline. Similarly, genetic associations near the TBL1XR1 gene with HbA1c levels were obtained in 438,069 European participants of the UK Biobank (Jurgens et al. 2023), whereas random glucose level associations were obtained in a different study of 476,326 individuals without diabetes of European (n=459,772) and other ancestries (n=16,554) (Lagou et al. 2023). The UK Biobank contributed the majority of the participants for these analyses.

We sought to look up the genetic associations for TBL1X and TBL1XR1 with β -cell function measurements, including HOMA-B, fasting insulin, and fasting proinsulin levels. Due to the lack of availability of those traits in the UK Biobank study and no genetic association summary statistics available for chromosome X, we were not able to obtain the genetic associations with the β -cell function measurements for TBL1X. As for TBL1XR1, no significant genetic associations were found with these measurements as proxies for hypoinsulinemia. This could potentially be due to the limited sample size for these traits (HOMA-B: n=46,186, fasting proinsulin: n=45,826), whereas there is more statistical power to detect significant genetic associations for HbA1c and random glucose given the availability of large sample sizes.

Although, as the reviewer pointed out, there is a lack of direct genetic association with β -cell function measurements in human cohorts due to the limited statistical power of those kinds of phenotypes, the significant genetic associations with HbA1c and random glucose levels indeed indicate that TBL1X/TBL1XR1 play a role in glucose homeostasis. The role in β -cells is further supported by our biological experiments.

We have added the detailed description of the cohorts used for genetic analyses to the methods description of the genetic association studies, line 1087-1092, and commented on the limitation of the genetic assessment in the discussion (line 656).

Minor points:

- Please check citation. Some citation does not match what's the author stated in the manuscript. e.g. Line 456-457 Pax6 depletion ~ 44 (44. Abe, Y. et al. RANK ligand ~). Isn't this citation 34.?

Indeed, we had some issues with endnote before submission, but have now double-checked and corrected all references throughout the manuscript.

- Figure 6a. Please provide statistic analyses.

Statistical analysis was added. Please see Fig. 6a and the respective figure legend for additional information.

References

Gutiérrez, Giselle Domínguez; Bender, Aaron S.; Cirulli, Vincenzo; Mastracci, Teresa L.; Kelly, Stephen M.; Tsigos, Aristotelis et al. (2017): Pancreatic β cell identity requires continual repression of non- β cell programs. In: *The Journal of clinical investigation* 127 (1), S. 244–259. DOI: 10.1172/JCI88017.

Hrovatin, Karin; Bastidas-Ponce, Aimée; Bakhti, Mostafa; Zappia, Luke; Büttner, Maren; Salinno, Ciro et al. (2023): Delineating mouse β -cell identity during lifetime and in diabetes with a single cell atlas. In: *Nature metabolism* 5 (9), S. 1615–1637. DOI: 10.1038/s42255-023-00876-x.

Jurgens, Sean J.; Pirruccello, James P.; Choi, Seung Hoan; Morrill, Valerie N.; Chaffin, Mark; Lubitz, Steven A. et al. (2023): Adjusting for common variant polygenic scores improves yield in rare variant association analyses. In: *Nature genetics* 55 (4), S. 544–548. DOI: 10.1038/s41588-023-01342-w.

Lagou, Vasiliki; Jiang, Longda; Ulrich, Anna; Zudina, Liudmila; González, Karla Sofia Gutiérrez; Balkhiyarova, Zhanna et al. (2023): GWAS of random glucose in 476,326 individuals provide insights into diabetes pathophysiology, complications and treatment stratification. In: *Nature genetics* 55 (9), S. 1448–1461. DOI: 10.1038/s41588-023-01462-3.

Matta, Leonardo; Weber, Peter; Erener, Suheda; Walth-Hummel, Alina; Hass, Daniela; Bühler, Lea K. et al. (2025): Chronic intermittent fasting impairs β cell maturation and function in adolescent mice. In: *Cell reports* 44 (2), S. 115225. DOI: 10.1016/j.celrep.2024.115225.

Olaniru, Oladapo Edward; Kadolsky, Ulrich; Kannambath, Shichina; Vaikkinen, Heli; Fung, Kathy; Dhama, Pawan; Persaud, Shanta J. (2023): Single-cell transcriptomic and spatial landscapes of the developing human pancreas. In: *Cell metabolism* 35 (1), 184-199.e5. DOI: 10.1016/j.cmet.2022.11.009.

Swisa, Avital; Avrahami, Dana; Eden, Noa; Zhang, Jia; Feleke, Eseye; Dahan, Tehila et al. (2017): PAX6 maintains β cell identity by repressing genes of alternative islet cell types. In: *The Journal of clinical investigation* 127 (1), S. 230–243. DOI: 10.1172/JCI88015.

Thisted, Louise; Østergaard, Mette V.; Pedersen, Annemarie A.; Pedersen, Philip J.; Lindsay, Ross T.; Murray, Andrew J. et al. (2020): Rat pancreatectomy combined with isoprenaline or uninephrectomy as models of diabetic cardiomyopathy or nephropathy. In: *Scientific reports* 10 (1), S. 16130. DOI: 10.1038/s41598-020-73046-8.

Yianni, Val; Sharpe, Paul T. (2022): Single Cell RNA-Seq: Cell Isolation and Data Analysis. In: *Methods in molecular biology (Clifton, N.J.)* 2403, S. 81–89. DOI: 10.1007/978-1-0716-1847-9_7.

Thank you for submitting your manuscript "TBL1X/TBL1XR1 are key regulators of a novel gene regulatory network controlling β -cell identity" to Nature Communications. I am delighted to say that we are happy, in principle, to publish it under an open access license.

Thank you, we are very happy that you have deemed our manuscript worthy of publication in Nature Communications.

We have now revised our paper according to the check list, as well as the remaining reviewer comments (see below). The corresponding author's ORCID ID is 0000-0003-3926-1534, and the MTS account is linked to ORCID.

REVIEWERS' COMMENTS

Reviewer #1 (Remarks to the Author):

Walth-Hummel et al. have very thoroughly addressed my concerns. I only have one minor comment that Figure S3C is not very legible and could be expanded so that the text is not overlapping.

We have changed that in the latest version.

Reviewer #2 (Remarks to the Author):

I thank the authors for their substantial efforts in revision. The added experiments and clarifications have improved the manuscript overall, including the new mouse knockdown analyses and the overexpression studies in both mouse islets and a human β -cell model. However, the central mechanistic question remains insufficiently resolved: do TBL1X/TBL1XR1 regulate β -cell function by directly controlling insulin gene expression (INS/Ins1/Ins2), rather than indirectly via secretion machinery, cell state changes, or mixed endocrine-cell effects?

1. Human knockdown: Insulin gene expression is still not directly quantified in the human β -cell model upon TBL1X/TBL1XR1 knockdown. Given the stated hypothesis, the authors should measure INS mRNA in EndoC- β H1 (or comparable human β -cell model, human islet) under knockdown, alongside secretion readouts.

2. Overexpression interpretation: While overexpression experiments are appreciated, the results differ between mouse islets and EndoC cells, and INS expression in EndoC overexpression conditions is not directly measured. Please quantify INS mRNA in EndoC overexpression experiments and reconcile any species/model differences (including the possibility of non- β endocrine contributions in intact mouse islets).

3. Human islet cell-type specificity: The scRNA-seq analysis is helpful, and mouse data suggest endocrine-wide expression, increasing the need for β -cell-autonomous evidence. For human islets, the presented data are not sufficiently clear to assess specificity. Please provide clearer human evidence using higher-quality datasets and more quantitative visualization (e.g., dot plots with % expressing and mean expression).

We thank the reviewer for the thorough assessment of our manuscript. Our data on insulin promoter activity support a direct transcriptional regulation of the insulin genes by TBL/R1.

However, as the reviewer points out, in some instances the effects on insulin secretion are stronger than the effects on insulin gene expression (e.g. upon TBL/R1 over expression, Fig. 5i-k). We cannot rule out that TBL/R1 affect additional mechanisms apart from insulin gene transcription that also contribute to the overall insulin secretion cascade, such as translation, secretion etc. Further, RNAseq data demonstrate reduced expression of β -cell identity genes (Pdx1, Nkx6.1 etc.) upon TBL/R1 depletion, which are transcription factors that also directly affect insulin gene expression.

It would need more detailed, dedicated studies in the future to unravel the relative contributions of direct / indirect transcriptional effects, as well as additional mechanisms including translation, protein degradation, secretion etc. to the overall phenotype of TBL/R1 defective β -cells. The relative contributions of the different insulin regulating mechanisms may differ between mouse and human systems.

We have now carefully revised the manuscript to highlight which effects relate directly to promoter activity, vs. gene expression (including in the abstract), and have further discussed a possible feedback mechanism with other β -cell specific transcription factors to highlight that TBL/R1 are not exclusive in regulating insulin gene transcription.

Reviewer #3 (Remarks to the Author):

The authors have adequately addressed my major concerns as outlined previous round or revision.

Overall, the revised manuscript presents a comprehensive and mechanistically supported study demonstrating an important role of TBL1X/TBL1XR1 in maintaining β -cell identity and function.

Minor suggestion

While the mechanistic interaction with PAX6 is convincingly demonstrated, the term “gene regulatory network” may slightly overstate the scope of the global regulatory evidence presented. The authors may consider modestly tempering this wording in discussion to better align with the scale of mechanistic data provided.

We thank the reviewer for this comment and have accordingly changed the wording in the discussion.